# Neonatally imprinted stromal cell subsets induce tolerogenic dendritic cells in mesenteric lymph nodes

Joern Pezoldt [1], Maria Pasztoi[1], Mangge Zou[1], Carolin Wiechers[1], Michael Beckstette[1], Guilhem R. Thierry[2], Ehsan Vafadarnejad[3], Stefan Floess[1], Panagiota Arampatzi [4], Manuela Buettner[5,6], Janina Schweer[7], Diana Fleissner[1], Marius Vital[8], Dietmar H. Pieper[8], Marijana Basic[6], Petra Dersch[7], Till Strowig [9], Mathias Hornef[10], André Bleich [6], Ulrike Bode[5], Oliver Pabst[11], Marc Bajénoff[2], Antoine-Emmanuel Saliba [3] & Jochen Huehn [1]

Gut-draining mesenteric lymph nodes (mLNs) are important for inducing peripheral tolerance towards food and commensal antigens by providing an optimal microenvironment for de novo generation of Foxp3[+] regulatory T cells (Tregs). We previously identified microbiota-imprinted mLN stromal cells as a critical component in tolerance induction. Here we show that this imprinting process already takes place in the neonatal phase, and renders the mLN stromal cell compartment resistant to inflammatory perturbations later in life. LN transplantation and single-cell RNA-seq uncover stably imprinted expression signatures in mLN fibroblastic stromal cells. Subsetting common stromal cells across gut-draining mLNs and skin-draining LNs further refine their location-specific immunomodulatory functions, such as subset-specific expression of *Aldh1a2/3*. Finally, we demonstrate that mLN stromal cells shape resident dendritic cells to attain high Treg-inducing capacity in a Bmp2-dependent manner. Thus, crosstalk between mLN stromal and resident dendritic cells provides a robust regulatory mechanism for the maintenance of intestinal tolerance.

[1] Department Experimental Immunology, Helmholtz Centre for Infection Research, 38124 Braunschweig, Germany. [2] CNRS, INSERM, CIML, Aix Marseille University, 13284 Marseille, France. [3] Helmholtz Institute for RNA-based Infection Research, 97080 Wuerzburg, Germany. [4] Core Unit Systems Medicine, University of Wuerzburg, 97080 Wuerzburg, Germany. [5] Institute of Functional and Applied Anatomy, Hannover Medical School, 30625 Hannover, Germany. [6] Institute for Laboratory Animal Science and Central Animal Facility, Hannover Medical School, 30625 Hannover, Germany. [7] Department Molecular Infection Biology, Helmholtz Centre for Infection Research, 38124 Braunschweig, Germany. [8] Research Group Microbial Interactions and Processes, Helmholtz Centre for Infection Research, 38124 Braunschweig, Germany. [9] Research Group Microbial Immune Regulation, Helmholtz Centre for Infection Research, 38124 Braunschweig, Germany. [10] Institute of Medical Microbiology, RWTH Aachen, 52074 Aachen, Germany. [11] Institute of Molecular Medicine, RWTH Aachen, 52074 Aachen, Germany. These authors contributed equally: Joern Pezoldt, Maria Pasztoi. Deceased: Ulrike Bode. Correspondence and requests for materials should be addressed to J.H. (email: Jochen.Huehn@helmholtz-hzi.de)

Immediately after birth, extensive colonization of body surfaces including the intestinal mucosa begins, a process that continues through childhood until a stable microbiota is established[1,2]. Microbiota contribute to the hosts' health by occupying niches for pathogenic microbes, breaking down indigestible material, synthesizing vitamins, and providing an important stimulus for the development and maturation of the mucosal and systemic immune system[1,3,4]. Multiple immune mechanisms are in place shortly after birth to tolerate the high tide of bacterial antigens, resulting in a temporarily limited overreaction of the immune system towards microbiota, but at the same time increasing the susceptibility of newborn infants to infection[1,5].

Foxp3$^+$ regulatory T cells (Tregs) play a central role in immune tolerance to self and foreign antigens[6]. The majority of Foxp3$^+$ Tregs are generated during thymic T cell development[7]. However, the peripheral Treg pool is expanded for specificities towards food antigens and microbiota by peripheral conversion of conventional Foxp3$^-$CD4$^+$ T cells into Foxp3$^+$ Tregs (peripherally-induced Tregs, pTregs)[8]. The generation of pTregs most efficiently takes place within gut-draining lymph nodes (LNs), including mesenteric LNs (mLNs)[9–11]. Subsequently, these pTregs migrate to and expand within the intestinal lamina propria[12], a process supported by microbiota and their metabolic products[13,14]. This division of labor between LNs as the initiation hub of Treg induction, and the intestinal lamina propria as the venue of expansion, requires stably compartmentalized cellular and molecular machinery.

An increasing body of evidence suggests that the unique tolerogenic properties of mLNs are not only shaped by tissue-derived migratory CD103$^+$ dendritic cells (DCs)[9,10], but also by resident LN stromal cells (SCs)[15], including fibroblastic stromal cells (FSCs) which dominate the T cell zone[15]. FSCs isolated from mLNs express high levels of the retinoic acid (RA)-synthesizing enzyme retinal aldehyde dehydrogenase 2 (Aldh1a2), and thereby contribute to the tolerogenic, Treg-inducing micromilieu[9,16]. Furthermore, FSCs are important for LN architecture by serving as structural scaffolds and by providing survival factors for migrating lymphocytes and DCs, thereby participating in the orchestration of appropriate cell–cell interactions required for the initiation of adaptive immune responses[15,17,18]. This intimate interaction suggests that FSCs are critically involved in the modulation of immune responses, best evidenced by the finding that they can attenuate acute inflammatory T cell responses[19]. However, little is known how FSC subsets functionally differ across LNs draining diverse tissues.

Reciprocal LN transplantations, where only graft-derived SCs are retained in the transplanted LN[16], provide valuable tools for analyzing the contribution of LN SCs to LN-specific immune responses. Using this technique we recently demonstrated that LN SCs critically contribute to the high Treg induction within mLNs and stably maintain their tolerogenic properties in a skin-draining environment[11]. Importantly, efficient Treg induction within mLNs relies on antigen presentation by DCs[11], and recent studies have proposed crosstalk between LN SCs and DCs[15,20,21], including the responsiveness of DCs to environmental cues derived from the SC compartment[15,16]. However, it is only incompletely understood which functional properties of DCs are educated by mLN SCs.

Here we report that mLN SCs are imprinted for a high Treg-inducing capacity soon after birth, and once established can stably maintain their tolerogenic potential subsequent to gastrointestinal infection and inflammation. Microbiota are critically required to imprint tolerogenic properties into mLN FSCs across different subsets, ensuring perpetuation of intestinal tolerance throughout life. Once imprinted, LN SCs can instruct LN-resident DCs (resDCs) to foster efficient Treg induction. Together, these observations reveal a mechanism whereby neonatally imprinted mLN SCs contribute to lifelong homeostatic intestinal tolerance by constantly modulating functional properties of resDCs.

## Results

**mLN SCs show inflammation-resistant tolerogenic properties.** We previously demonstrated that mLN SCs contribute to the high Treg-inducing properties of mLNs[11]. To collect further evidence for the stability of the tolerogenic phenotype of mLN SCs, mLNs were transplanted into the popliteal fossa of recipient mice after excision of the endogenous popliteal LN. Skin-draining LNs (pLNs) were transplanted as controls. Fifty weeks after transplantation, the Treg-inducing capacity of transplanted mLNs was analyzed by adoptive transfer of T cell receptor (TCR)-transgenic, ovalbumin (Ova)-specific naive Foxp3$^-$CD4$^+$ T cells, followed by systemic intravenous application of Ova peptide (Fig. 1a). At day 3 after antigen application, flow cytometric analysis revealed a comparable proliferation of Ova-specific T cells within transplanted pLNs and mLNs (Fig. 1b). Strikingly, the high Treg-inducing capacity of mLNs was maintained fifty weeks after transplantation into the popliteal fossa (Fig. 1b-c), demonstrating that mLN SCs can stably and durably maintain their tolerogenic properties in a skin-draining environment.

Next, we investigated whether the tolerogenic properties of mLN SCs can resist inflammatory perturbations within the intestine. First, mice were orally infected with enteropathogenic *Yersinia pseudotuberculosis*, a gastrointestinal pathogen known to enter mLNs shortly after infection (Supplementary Fig. 1A). As expected, *Yersinia* infection resulted in profound changes of the mLN SC compartment. At day 3 post infection (p.i.), the number of CD45$^-$CD24$^-$gp38$^+$CD31$^-$ FSCs was significantly reduced compared to uninfected controls, and FSCs displayed an activated phenotype with increased MHCII expression (Supplementary Fig. 1B–C). Four weeks p.i., a time point when *Yersiniae* were cleared from mLNs (Supplementary Fig. 1A), the number of FSCs was significantly increased, and the FSCs still showed an activated phenotype (Supplementary Fig. 1B–C), suggesting that the FSCs had significantly proliferated in response to the infection. To assess whether infection-induced changes to the mLN SC compartment can persistently alter the high Treg-inducing capacity of mLNs, we transplanted mLNs of mice four weeks p.i. with *Yersinia* into the popliteal fossa of uninfected recipient mice. Eight to ten weeks later the Treg-inducing capacity of transplanted mLNs was analyzed as described above, so that any impact of previous infection on the frequency of de novo induced Foxp3$^+$ Tregs could be observed (Supplementary Fig. 1D). This analysis indicated that the observed infection-induced changes to the mLN SC compartment did not persistently alter the high Treg-inducing capacity of mLNs. In a second approach, we utilized the chronic dextran sodium sulfate (DSS) colitis model to study whether a chronic inflammatory perturbation could abrogate the high Treg-inducing properties of mLN SCs. After four cycles of DSS treatment (Fig. 1d), when mice had developed a chronic colitis as indicated by a significant shortening of colon length, as well as increased spleen size (Fig. 1e), mLNs and LNs draining the caecum and proximal colon (caeLNs) were transplanted into the popliteal fossa of recipient mice as described above. Interestingly, eight to ten weeks after transplantation, both caeLNs and mLNs still showed a high Treg-inducing capacity (Fig. 1f). Together, these results highlight the stability of the tolerogenic properties of mLN SCs, by withstanding acute and even chronic inflammatory perturbations.

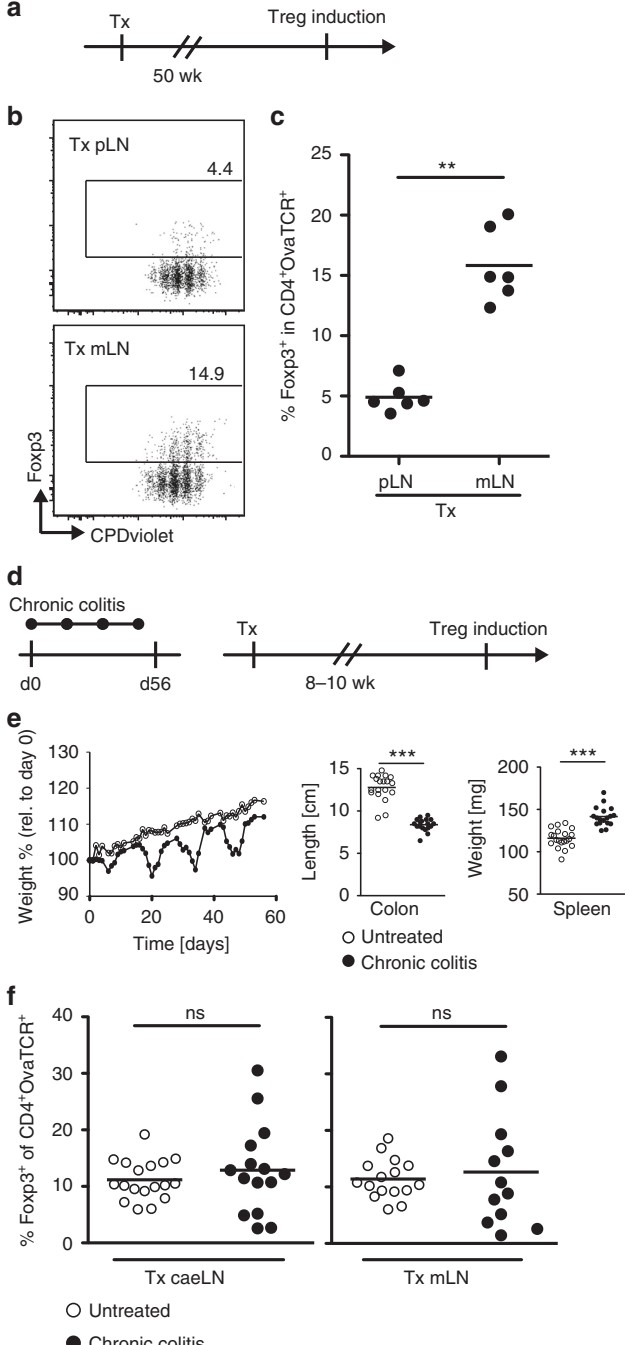

**Fig. 1** Long-lasting, inflammation-resistant retention of tolerogenic properties in transplanted mLNs. **a-c** mLNs or pLNs from SPF-housed mice were transplanted into the popliteal fossa of SPF-housed recipient mice. Fifty weeks later, transplanted mice received CPDviolet-labeled cells isolated from Foxp3$^{hCD2}$xRag2$^{-/-}$xDO11.10 mice. On two consecutive days, recipients were immunized via repetitive *i.v.* injection of Ova$_{323-339}$ peptide and analyzed on day 3 after the first immunization. **b** Exemplary dotplots depict de novo Treg induction in indicated transplanted LNs gated on adoptively transferred OvaTCR$^+$CD4$^+$ cells. **c** Scatterplot summarizes frequencies of de novo induced Foxp3$^+$ Tregs among transferred OvaTCR$^+$CD4$^+$ cells recovered from transplanted LNs. Data pooled from two independent experiments are shown ($n = 6$). **d–f** SPF-housed BALB/c mice aged eight weeks were treated with 5% DSS in drinking water ad libitum for four days followed by ten days of normal drinking water. This treatment cycle was repeated four times. **e** Relative changes in body weight during DSS-induced chronic colitis were monitored over time (filled circles). Uninfected mice served as controls (open circles). Scatterplots summarize colon length and spleen weight of DSS-treated mice and untreated controls. **f** caeLNs or mLNs from DSS-treated mice (chronic colitis) and from untreated controls were transplanted to the popliteal fossa of healthy mice after removal of the endogenous popliteal LN. After eight to ten weeks of reconstitution, de novo Treg induction was assessed in transplanted mLNs. Scatterplots summarize frequencies of de novo induced Foxp3$^+$ Tregs among transferred OvaTCR$^+$CD4$^+$ cells recovered from transplanted mLNs (open circles, untreated; filled circles, chronic colitis). Data pooled from four independent experiments are shown ($n = 12$–18). caeLN caecum-draining and colon-draining lymph node, DSS dextran sodium sulfate, rel. relative, Tx transplantation

taken from 24 and 60 day-old mice (Fig. 2a). Thus, stable imprinting of tolerogenic properties within mLN SCs occurs very early during ontogeny in the neonatal period, when commensal colonization of body surfaces starts[1,2].

**Microbiota imprint tolerogenic properties into mLN SCs**. Our previous findings indicated that microbiota might contribute to the durable imprinting of tolerogenic properties in mLN SCs, as transplanted mLNs taken from germ-free (GF) mice did not show any increased Treg-inducing capacity[11]. To further underpin the role of microbiota in this imprinting process, GF mice aged three to four weeks or seven to eight weeks were colonized by co-housing with specific pathogen-free (SPF) mice. Subsequently, mLNs from co-housed GF mice were transplanted into the popliteal fossa of recipient mice. Eight to twelve weeks after transplantation, the Treg-inducing capacity of transplanted LNs was analyzed as described before. Interestingly, when we determined the frequency of de novo induced Foxp3$^+$ Tregs, no difference between transplanted mLNs taken from co-housed GF mice and transplanted mLNs of age-matched SPF controls (Fig. 2b) was observed. These data indicate that microbiota are sufficient to stably imprint tolerogenic properties within mLN SCs.

**Intact mLN SC tolerogenic property after microbiota changes**. Having confirmed that microbiota can permanently shape the immunomodulatory capacity of mLN SCs, we next asked whether global alteration of the microbiota composition during the critical neonatal period might influence this imprinting process. Therefore, we treated pregnant mothers starting from embryonic day 7 and their offspring with various antibiotics, namely streptomycin, polymyxin B and vancomycin, all targeting different classes of bacteria (Fig. 2c). As expected, these treatments led to alterations of the microbiota composition, with vancomycin causing the

**mLN SCs acquire tolerogenic properties rapidly after birth**. To define when SCs attain their stable, transplantation-resistant and inflammation-resistant functions, we transplanted mLNs of neonatal, 10, 24, and 60 day-old mice into the popliteal fossa of adult recipient mice. Successful engraftment of neonatal mLNs was verified by transplanting neonatal mLNs of β-actin enhanced cyan-fluorescent protein (eCFP) reporter mice and demonstrating eCFP expression in FSCs re-isolated from transplanted mLNs (Supplementary Fig. 2A–B). Eight to twelve weeks after transplantation, the Treg-inducing capacity of transplanted LNs was analyzed as described before. Interestingly, transplanted neonatal mLNs showed a low Treg-inducing capacity (Fig. 2a), whereas mLNs from 10 day-old mice had already acquired a high Treg-inducing capacity, and no significant further increase in the frequency of induced Tregs was observed in transplanted mLNs

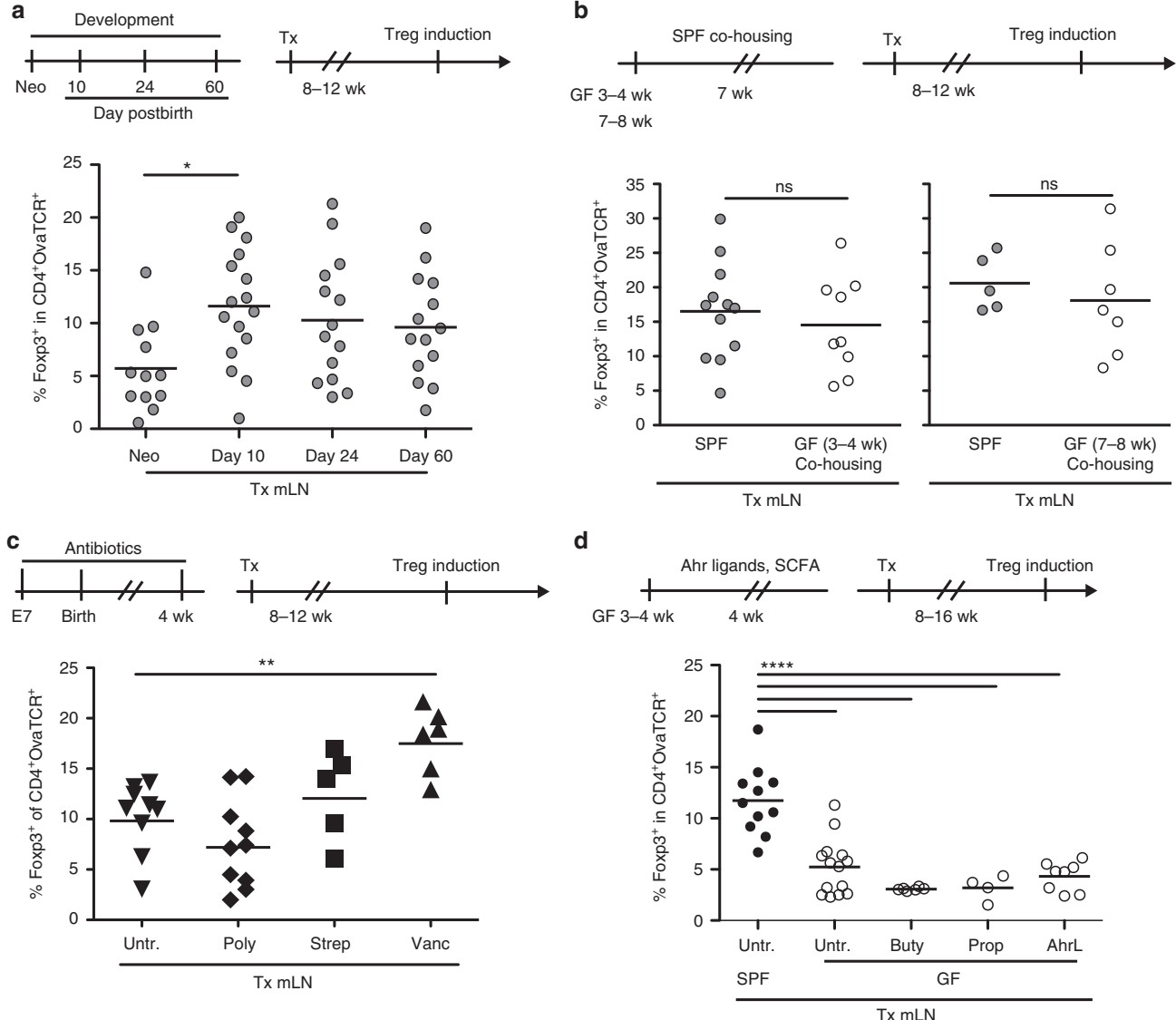

**Fig. 2** Microbiota trigger imprinting of tolerogenic properties into mLN SCs early after birth. Indicated LNs were transplanted into the popliteal fossa of SPF-housed recipient mice. Eight to sixteen weeks later, transplanted mice received CPDviolet-labeled cells isolated from Foxp3^hCD2xRag2^−/−xDO11.10 mice. On two consecutive days, recipients were immunized via repetitive i.v. injection of Ova_{323-339} peptide and analyzed on day 3 after the first immunization. **a** mLNs of neonatal, 10, 24, and 60 days old SPF-housed mice were transplanted. Scatterplot summarizes frequencies of de novo induced Foxp3^+ Tregs among transferred OvaTCR^+CD4^+ cells recovered from transplanted mLNs. Data pooled from four independent experiments are shown ($n = 12$–14). **b** GF mice of indicated age were co-housed with age-matched SPF mice for seven weeks, followed by mLN transplantation. Scatterplot summarizes frequencies of de novo induced Foxp3^+ Tregs among transferred OvaTCR^+CD4^+ cells recovered from transplanted LNs. Data pooled from three to four independent experiments are shown ($n = 5$–12). **c** Mothers and their offspring were treated with antibiotics starting seven days post conception until four weeks of age, followed by mLN transplantation. Untreated mice served as controls. Scatterplot summarizes frequencies of de novo induced Foxp3^+ Tregs among transferred OvaTCR^+CD4^+ cells recovered from transplanted mLNs. Data pooled from two independent experiments are shown ($n = 5$–10). **d** Three to five week old GF mice were treated orally via drinking water with either propionate or butyrate at 100 mM or a mixture of AhR ligands including IAA, I3C, and ITE at 0.1 mM. mLNs of untreated and treated GF mice were transplanted, and mLNs from SPF mice served as additional controls. Scatterplot summarizes frequencies of de novo induced Foxp3^+ Tregs among transferred OvaTCR^+CD4^+ cells recovered from transplanted mLNs. Data pooled from two to four independent experiments ($n = 4$–14). Ahr aryl hydrocarbon receptor, buty butyrate, IAA indole-3-acetic acid, I3C indole-3-carboxaldehyde, ITE 2-(1H-indole-3-ylcarbonyl)-4-thiazol ecarboxylic acid methyl ester, neo neonatal, poly polymyxin B, prop propionate, strep streptomycin, Tx transplantation, untr. untreated, vanc vancomycin

most drastic changes and resulting in an elevated proportion of *Lactobacillaceae* as Gram-positive *Lactobacillus spp.* are commonly vancomycin-resistant (Supplementary Fig. 2C–D). Next, mLNs of antibiotic-treated offspring were transplanted into the popliteal fossa of untreated recipient mice, followed by assessment of their Treg-inducing capacities as described before. None of the treatment regimens abrogated the tolerogenic properties of

mLN SCs. Remarkably, vancomycin, which caused the strongest reduction in microbiota diversity, actually resulted in enhanced Treg induction within transplanted mLNs (Fig. 2c, Supplementary Fig. 2E).

Since our data indicated that different microbiota compositions are capable of imprinting tolerogenic properties within mLN SCs, albeit to different degrees, we next inquired which bacteria-

derived molecular signals were required for imprinting. Intestinal microbiota provide different metabolites, and particularly short-chain fatty acids (SCFA) and aryl-hydrocarbon receptor (Ahr) ligands which have been shown to contribute to lasting alterations of immune homeostasis[13,14,22]. Thus, we here assessed whether these ligands are sufficient to imprint high Treg-inducing capacity into mLN SCs. GF mice were orally treated for four weeks with either propionate, butyrate or a mixture of endogenous Ahr ligands, namely 2-(1H-indole-3-ylcarbonyl)-4-thiazolecarboxylic acid methyl ester (ITE), indole-3-acetic acid (IAA) and indole-3-carboxaldehyde (I3C). mLNs of treated GF mice were transplanted into the popliteal fossa of untreated SPF mice, followed by assessment of their Treg-inducing capacities as described before. Neither SCFAs nor Ahr ligands were sufficient to imprint tolerogenic properties into mLN SCs (Fig. 2d). In summary, our findings indicate that microbiota can imprint tolerogenic properties within mLN SCs independent of their composition and major immunomodulatory metabolites.

**Microbiota imprint tissue-specific properties into mLN FSCs.** Next, we aimed to unravel the impact of microbiota on mLN SCs at the molecular level. Previous studies provided comparative microarray analyses of FSCs isolated from pLNs and mLNs revealing differential expression of several genes[23,24]. However, these studies did not investigate if microbiota influence the LN-specific transcriptional signatures of FSCs. Thus, we isolated FSCs from mLNs and pLNs of both SPF and GF mice. FSCs were subjected to transcriptome analyses by RNA-seq. Surprisingly, global inspection of these data by hierarchical clustering of differentially expressed genes (DEGs, $|\log_2(FC)| \geq 1$, $q$-value $\leq 0.05$) revealed that colonization status scarcely impacted the transcriptomes of FSCs from pLNs and mLNs alike (Fig. 3a). A direct comparison of all four conditions underlined the impact of anatomic localization on the transcriptional signatures of FSCs (Supplementary Fig. 3A). Gene ontology (GO) analysis revealed that key biological processes including RA metabolism were maintained independent of microbial colonization (Supplementary Fig. 3B, Supplementary Table 1).

Having demonstrated that colonization status barely affects the transcriptome of FSCs from mLNs, we next sought to unravel whether microbiota could be instrumental in stabilizing the mLN-specific transcriptional signature. Accordingly, we transplanted mLN-SPF or mLN-GF to the skin-draining popliteal fossa of SPF recipient mice, while pLN-SPF were transplanted as a surgical control. Eight to fifteen weeks later, FSCs were isolated from transplanted LNs and subjected to low-input RNA-seq (RNA-seq^L), with FSCs isolated from endogenous LNs as additional controls. Interestingly, FSCs from transplanted mLN-SPF maintained a substantial fraction of the mLN-specific transcriptional signature, with 108 and 368 genes showing persistent up-regulation and down-regulation, respectively (Fig. 3b left, Supplementary Table 2). In contrast, FSCs from transplanted mLN-GF were strongly influenced by the skin microenvironment and almost completely adapted to the transcriptional signature of FSCs from pLNs (Fig. 3b right). With 32 up-regulated and 49 down-regulated genes only a small fraction of the mLN-specific transcriptional signature was maintained (Fig. 3b right, Supplementary Fig. 3C). Together, these data suggest that microbiota are essential for stabilizing the mLN-specific transcriptional signature within FSCs.

GO analysis of those DEGs maintained in FSCs from transplanted mLN-SPF revealed a permanent repression of inflammation-associated biological pathways, while RA biosynthesis remained stably up-regulated (Fig. 3c). Remarkably, *Aldh1a2*, an enzyme responsible for RA synthesis, was persistently

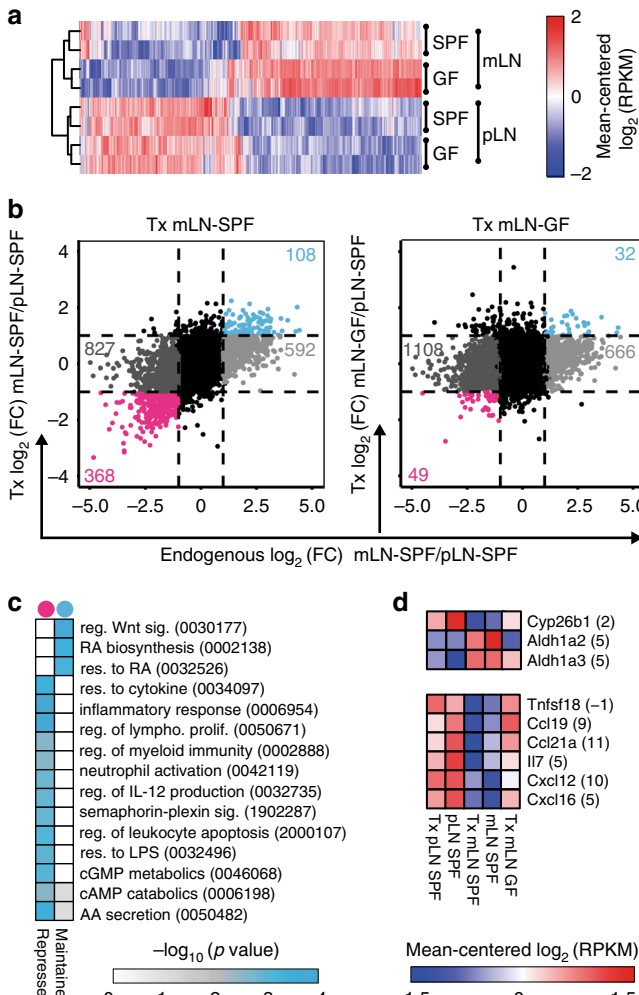

**Fig. 3** Location-specific transcriptional signatures of mLN FSCs are imprinted by microbiota. **a** CD45^−Ter119^−CD31^−gp38^+ FSCs were isolated from mLNs and pLNs of GF or SPF mice, and RNA-seq and subsequent analysis was performed. DEGs were identified in colonization-dependent (SPF vs. GF) and location-dependent (mLNs vs. pLNs) pairwise comparisons ($|\log_2(FC)| \geq 1$ and $q$-value $\leq 0.05$). Heatmap represents 1411 DEGs. Mean-centered $\log_2(RPKM)$ values are depicted. Data pooled from two independent experiments. **b** CD45^−CD24^−CD31^−gp38^+ FSCs were isolated from endogenous mLN-SPF and pLN-SPF, transplanted pLN-SPF and mLN-SPF or transplanted mLN-GF. RNA-seq^L and subsequent analysis was performed. Colored numbers in scatterplots represent DEGs ($|\log_2(FC)| \geq 1$ and $q$-value $\leq 0.05$) for the pair-wise comparisons of FSCs. On the x-axis $\log_2(FC)$ of gene expression from FSCs (endogenous LNs) is plotted. (left) On the y-axis $\log_2(FC)$ of gene expression from FSCs from transplanted mLN-SPF vs. pLN-SPF is plotted. (right) On the y-axis $\log_2(FC)$ of gene expression from FSCs from transplanted mLN-GF vs. pLN-SPF is plotted. Data pooled from two to four independent experiments ($n = 3$–8). **c** GO analysis of biological processes of genes persistently up-regulated (Maintained, blue dot) or down-regulated (Repressed, pink dot) in FSCs of transplanted mLN-SPF. **d** Heatmaps of expression of genes involved in RA metabolic process or of soluble mediators within FSCs from indicated groups. Numbers in brackets indicate average $\log_2(RPKM)$ expression of respective genes across all experimental groups. Data pooled from two to four independent experiments ($n = 3$–8). DEG differentially expressed genes, AA amino acid, FC fold-change, FSC fibroblastic stromal cells, GO gene ontology, RA retinoic acid, reg. regulation, res. response, RPKM reads per kilobase of exon length per million mapped reads, sig. signaling

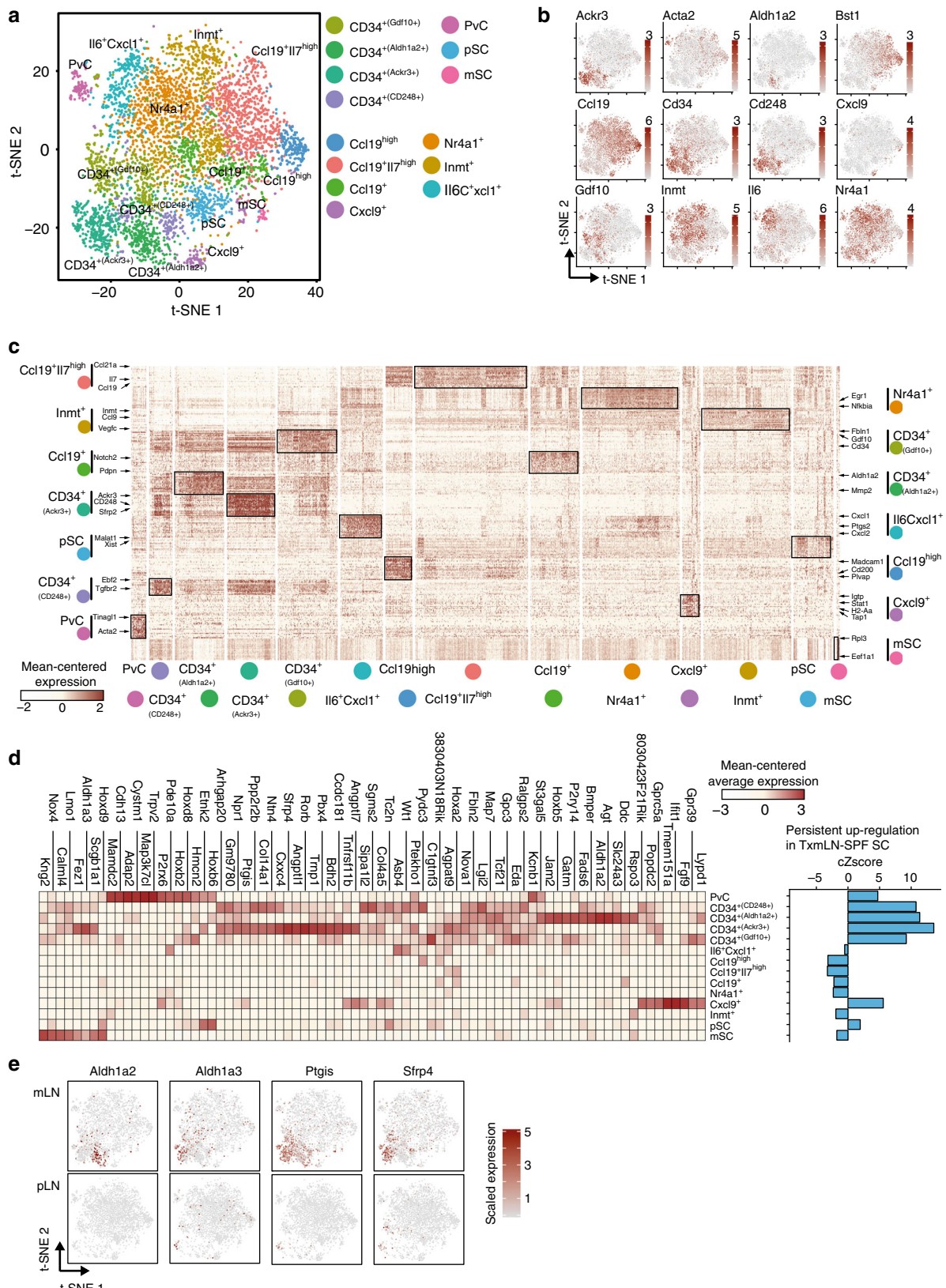

maintained only in FSCs from transplanted mLN-SPF but not in FSCs from transplanted mLN-GF, while the RA-degrading enzyme *Cyp26b1* remained repressed (Fig. 3d). Several important soluble mediators (*Tnfsf18*, *Cxcl12*, *Cxcl16*, *Il7*, *Ccl19*, *Ccl21a*) were maintained at lower levels in FSCs from transplanted mLN-SPF, but showed increased expression levels in FSCs from transplanted mLN-GF (Fig. 3d). In summary, transcriptome analysis of FSCs isolated from transplanted mLNs confirmed that the stabilization of a substantial fraction of the location-specific transcriptional signature depends on microbiota.

**Fig. 4** Location-dependent features are imprinted within selected SC subsets. Single cell suspensions from mLNs and pLNs were sorted for CD24⁻CD45⁻ cells and subjected to scRNA-seq. Non-endothelial SCs were identified as non-LECs, non-BECs, and $Pecam1^-$ and $Ackr4^-$. Data pooled from two independent experiments. **a** t-SNE plot of merged pLN SCs and mLN SCs showing cluster segregation. **b** Expression of subset-defining DEGs across all non-endothelial SCs on t-SNE plot. **c** Heatmap of 272 DEGs (up to 20 per cluster) with the most significant differential expression (minimal FDR-adjusted p-value). Marker genes for respective clusters are indicated with arrows. **d** Heatmap (left) depicts average expression per cluster of genes persistently up-regulated in SCs from transplanted mLN-SPF. Bargraph (right) shows cumulative Z-score (cZscore) of average expression per cluster for the genes persistently up-regulated in SCs from transplanted mLN-SPF. **e** Expression of indicated genes persistently up-regulated in non-endothelial SCs from transplanted mLN-SPF on t-SNE plot for mLN-SPF and pLN-SPF scRNA-seq. BEC blood endothelial cell, DEG differentially expressed gene, mSC metabolic stromal cells, pSC proliferating stromal cells, PvC perivascular cells, SC stromal cell, LEC lymphatic endothelial cell, t-SNE t-distributed stochastic neighbor embedding

## Location-dependent features are imprinted within SC subsets.

The complex anatomical infrastructure of the LN requires a separation of functional properties within SC subsets[15]. However, the precise role of FSC subsets cannot be resolved by transcriptional profiling of bulk FSCs. To get a fair picture of the SC subset composition within pLNs and mLNs of adult SPF mice and gain preliminary insights into the functional properties of these subsets, we performed single-cell RNA-seq (scRNA-seq) of the entire CD24⁻CD45⁻ cell population and acquired the transcriptomes of quality-controlled 3928 and 5329 single-cells for mLNs and pLNs, respectively. For integrative analysis of scRNA-seq of mLN SCs and pLN SCs we implemented sample alignment based on diagonal canonical correlation analysis to overlay the transcriptomes of both LNs and performed unsupervised clustering using t-SNE (Supplementary Fig. 4A)[25]. At first glance, we recognized known major LN SC subpopulations, including lymphatic endothelial cells (LEC, $Pdpn^+Pecam1^+$), blood endothelial cells (BEC, $Pdpn^-Pecam1^+$) and non-endothelial SCs (SC, $Pecam1^-Ackr4^-$), with the latter displaying the most pronounced heterogeneity. Importantly, Pdpn and Pecam1 expression alone were insufficient to separate LECs, BECs and non-endothelial SC at a single-cell level, although sufficient to distinguish cellular clusters based on the averaged expression (Supplementary Fig. 4A).

To get an unbiased picture of SC subsets within pLNs and mLNs, we aligned 2786 mLN SCs and the identical number of randomly sampled pLN SCs, while omitting all LECs and BECs (Fig. 4a). Fourteen transcriptional clusters harboring unique functional profiles were identified based on DEGs and GO analysis (Fig. 4a–b, Supplementary Fig. 4A–C). Importantly, the vast majority of clusters were found in both pLNs and mLNs (Supplementary Fig. 4D), suggesting that LNs are composed of similar SC subsets. We could recapitulate known SC subsets based on gene expression patterns, namely $Acta2^+$ pericytes (PvC) and $Ccl19^+$ T cell zone reticular cells (TRC) (Fig. 4a–b)[15,26]. Among the TRCs we were able to distinguish four subsets, namely Cxcl9⁺, Ccl19^high, Ccl19⁺ and Ccl19⁺Il7^high TRCs (Fig. 4a–c). Additionally, a subset expressing a variety of inflammatory and chemotactic mediators including Il6, Cxcl1, Ccl2, Ccl7, and Cxcl2, here termed Il6⁺Cxcl1⁺ SCs, was identified (Fig. 4a–c, Supplementary Table 3). We also detected subsets with higher, but not exclusive expression for Inmt and Nr4a1, previously termed Inmt⁺ and Nr4a1⁺ SCs, respectively[27]. Remarkably, a large population of CD34⁺ SCs, known to predominantly locate at the LN capsule and around larger vessels[28], was identified that could further be subdivided into four distinct subsets, namely CD34⁺(Aldh1a2+), CD34⁺(Ackr3+), CD34⁺(Gdf10+) and CD34⁺(Cd248+) SCs, each bearing distinct transcriptional profiles (Fig. 4a–c, Supplementary Fig. 4C, Supplementary Table 3). Finally, two subsets enriched for proliferating (pSC) and metabolically active (mSC) were detected (Fig. 4a, Supplementary Fig. 4c, Supplementary Table 3). By projecting transcriptional signatures from mLN SC clusters to pLN SC clusters using scmap[29], the majority of SC subsets were

reflected (Supplementary Fig. 4D). This finding is underscored by the overlap of the top 10 DEGs per cluster (Supplementary Fig. 4E, Supplementary Table 4 and 5), demonstrating a substantial conservation of LN SC subsets across different LNs.

Having identified novel non-endothelial SC subsets, we next asked if any one of these subsets could mainly account for the stably imprinted, mLN-specific transcriptional signature discovered within the total population of FSCs isolated from transplanted mLN-SPF (Fig. 3b–d, Supplementary Table 2). Thus, we calculated the cumulative Z-score (cZscore) for the 108 persistently up-regulated and 368 persistently down-regulated genes across all newly identified SC clusters identified by scRNA-seq (Fig. 4d). Interestingly, defined sets of the persistently up-regulated genes were mainly expressed by the four CD34⁺ SC subsets, as indicated by a cZscore > 0, but also in Cxcl9⁺ TRCs and PvCs (Fig. 4d). A similar picture was observed for the persistently down-regulated genes (cZscore < 0) (Supplementary Fig. 4f). Interestingly, the two RA-synthesizing enzymes Aldh1a2 and Aldh1a3 showed a mutually exclusive expression within two distinct CD34⁺ SC subsets, while Ptgis, an enzyme involved in cyclic prostaglandin synthesis, was expressed consistently across three out of the four CD34⁺ SC subsets (Fig. 4e). In order to identify the location of Aldh1a2-expressing non-endothelial SCs, we performed immunostaining and confocal imaging on tissue-sections of mLN-SPF and pLN-SPF from chimeric mice, which were generated by reconstituting $Ccl19^{Cre}$x$Rosa26^{tdT}$ individuals[30,31] with bone marrow from $Zbtb46^{GFP}$ reporter mice[32], allowing us to differentiate Ccl19-dependently fate-mapped SCs and Zbtb46-expressing DCs (Supplementary Fig. 5A–C). As expected, Aldh1a1/2⁺ DCs were abundantly found in mLNs but not pLNs, whereas no Aldh1a1/2-expressing SCs could be detected within the T cell zone (Supplementary Fig. 5B). It should be noted that a reliable identification of Aldh1a2-expressing SCs within the capsular region and in the vicinity of the medullary cords was prevented by unspecific background signals. In summary, the scRNA-seq data suggest that selected immunomodulatory properties of mLN SCs are confined to distinct SC subsets, preferentially of the CD34⁺ SC group.

## mLN SCs modulate resident DC composition and transcriptome.

Emerging evidence suggests that LN SCs and DCs closely interact[15,33,34]. Due to the location of FSCs within the T cell zone of the LN, and the previously reported superior Treg-inducing capacity of DCs from mLNs[9–11], we here aimed to determine the direct modulation of DCs by LN SCs. Thus, we transplanted mLNs and pLNs from adult SPF mice to the skin-draining popliteal fossa of SPF recipient mice. Eight to sixteen weeks later, we first assessed the composition of CD11c^hiMHCII⁺ resident DC (resDC) and CD11c⁺MHCII^hi migratory DC (migDC) subsets in transplanted LNs (Fig. 5a). It is important to note that in the LN transplant setting all resDCs and migDCs of donor origin have been replaced by recipient-derived DCs at the

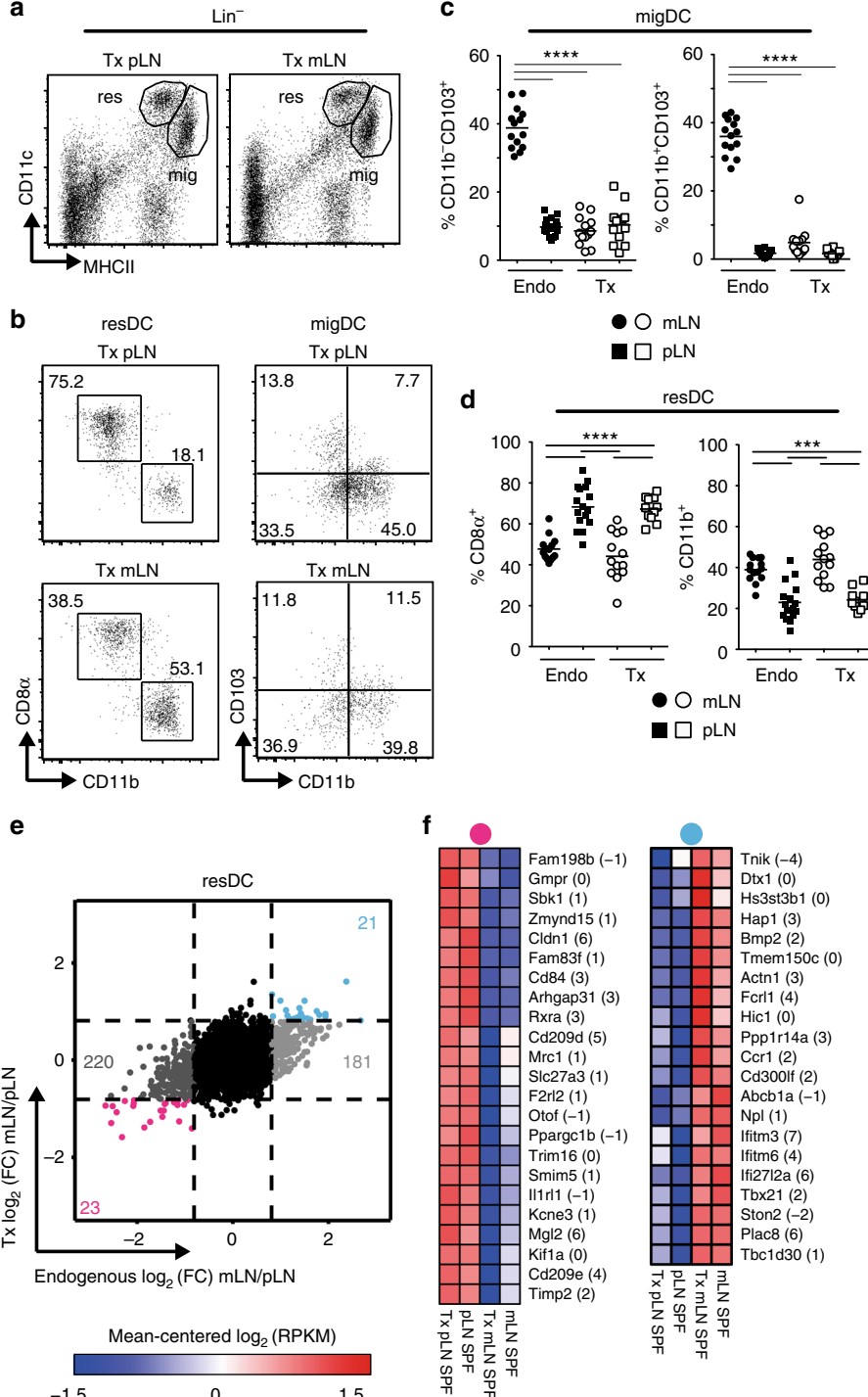

**Fig. 5** mLN SCs modulate subset composition and transcriptional signature of resident DCs. Indicated LNs were transplanted to the popliteal fossa of SPF-housed mice. Eight to sixteen weeks later, single cell suspensions were generated by enzymatic digestion. **a–d** Single cell suspensions were analyzed using flow cytometry. **a** Exemplary dotplot of viable Lin⁻ cells from transplanted LN. (**b**) Exemplary dotplot of subsets from resDCs and migDCs from indicated LNs. **c–d** Scatterplot shows frequencies of indicated subsets among migDCs (**c**) and resDCs (**d**); data pooled from two to three independent experiments ($n = 12$–16). **e** resDCs were isolated from indicated LNs. RNA-seq$^L$ and subsequent analysis was performed. Colored numbers in scatterplots represent DEGs for the respective pair-wise comparisons of resDCs ($|\log_2(FC)| \geq 1$ and $q$-value $\leq 0.05$). On the $x$-axis $\log_2 FC$ of gene expression from resDCs (endogenous LNs) is plotted. On the $y$-axis $\log_2(FC)$ of gene expression from resDCs of transplanted mLN-SPF vs. pLN-SPF is plotted. **f** Heatmap of expression of genes persistently down- (pink dot, left) or up-regulated (blue dot, right) in resDCs isolated from transplanted mLN-SPF. Numbers in brackets indicate average $\log_2(RPKM)$ expression of respective genes across all experimental groups. Data pooled from two to three independent experiments ($n = 3$–8). migDC migratory dendritic cell (Lin⁻CD11c⁺MHCII$^{high}$), resDC resident dendritic cell (Lin⁻CD11c$^{high}$MHCII⁺), Lin CD3⁺CD45R⁺Ly6G⁺F4/80$^{high}$, FC fold-change, RPKM reads per kilobase of exon length per million mapped reads, Endo endogenous, Tx transplanted

time of analysis (Supplementary Fig. 6A). As expected, a low frequency of CD103[+] migDC subsets was observed in both transplanted pLNs and mLNs (Fig. 5b–c), closely resembling the DC subset composition in endogenous pLNs (Supplementary Fig. 6B). These data suggest that the composition of migDCs is mainly influenced by the skin microenvironment, independent of the origin of the transplanted LN. In contrast, resDCs maintained their mLN-specific subset composition (Supplementary Fig. 6B) within transplanted mLNs (Fig. 5b and d), demonstrating that mLN SCs can modulate resDCs at the compartment level.

To elucidate the modulation of resDCs and migDCs by mLN SCs at the molecular level, we isolated resDCs and migDCs from transplanted mLNs and pLNs eight to sixteen weeks after transplantation and performed transcriptome analysis by RNA-seq[L]. Interestingly, for both resDCs and migDCs a fraction of ~10% of the total location-dependent DEGs ($\log_2 FC = 0.8$, $q$-value < 0.05) was stably maintained in transplanted mLNs (Fig. 5e, Supplementary Fig. 6C). migDCs retained the expression of characteristic marker genes, such as *Ccr7*, *Arc*, and *Irf4*[35–37], as well as key biological pathways (Supplementary Fig. 6D and E). Interestingly, the skin DC marker genes *Cd209*, *Cldn1*, *Mgl2*, and *Cd84*[38–41] remained stably down-regulated in resDCs from transplanted mLNs (Fig. 5f). Together, these data demonstrate that mLN SCs can modulate both migDCs and resDCs at the molecular level, thereby shaping their immunomodulatory properties.

**resDCs from transplanted mLNs retain Treg-inducing capacity.** Finally, we wanted to assess the Treg-inducing capacity of resDCs and migDCs shaped by mLN SCs. First, we ruled out any direct impact of mLN SCs on naïve T cells during de novo Treg induction. In vitro co-cultivation of polyclonally activated naïve T cells with CD24[−]CD45[−]gp38[+]CD31[−] FSCs isolated from mLNs was unaccompanied by an increased frequency of de novo induced Foxp3[+] Tregs when compared to co-cultures with FSCs isolated from pLN (Supplementary Fig. 7A). The lack of any direct impact of FSCs on Treg induction was further corroborated in the context of TGFβ1-dependent Treg induction in conjunction with blocking Timd4, a surface molecule predominantly expressed by pLN FSCs and described to inhibit Treg induction[42], further suggesting that DCs are required to confer FSC-dependent immune modulation (Supplementary Fig. 7A).

Hence, we isolated resDCs and migDCs from transplanted LNs eight to sixteen weeks after transplantation, while resDCs, as well as migDCs isolated from endogenous LNs were taken as controls. All DCs were cultured with Ova-TCR transgenic naïve T cells in the presence of Ova peptide, and the frequency of de novo induced Foxp3[+] Tregs was analyzed on day 5. In all conditions tested, a comparable proliferation was observed (Fig. 6a). The high Treg-inducing capacity of migDCs from endogenous mLNs was not maintained in migDCs isolated from transplanted mLNs (Fig. 6a–b). In contrast, resDCs from transplanted mLNs fully attained their higher Treg-inducing capacity (Fig. 6a and c). Interestingly, GO analysis of the mLN SC-dependent modulation of resDCs pointed towards genes involved in bone mineralization, including up-regulation of *Bmp2* encoding the bone morphogenic protein 2, whereas *Tgfb1* and *Tgfb3* were expressed at equal levels in resDCs from pLNs and mLNs (Fig. 6d, e). *Bmp2* belongs to the TGF-β superfamily and promotes TGFβ1-dependent Treg induction solely in a synergistic manner[43]. When culturing resDCs from mLNs and pLNs in the presence of the Bmp-antagonist Noggin, mLN resDC-mediated Treg induction was significantly reduced (Fig. 6f), whereas exogenous supplementation with Bmp2 resulted in an increased frequency of de novo

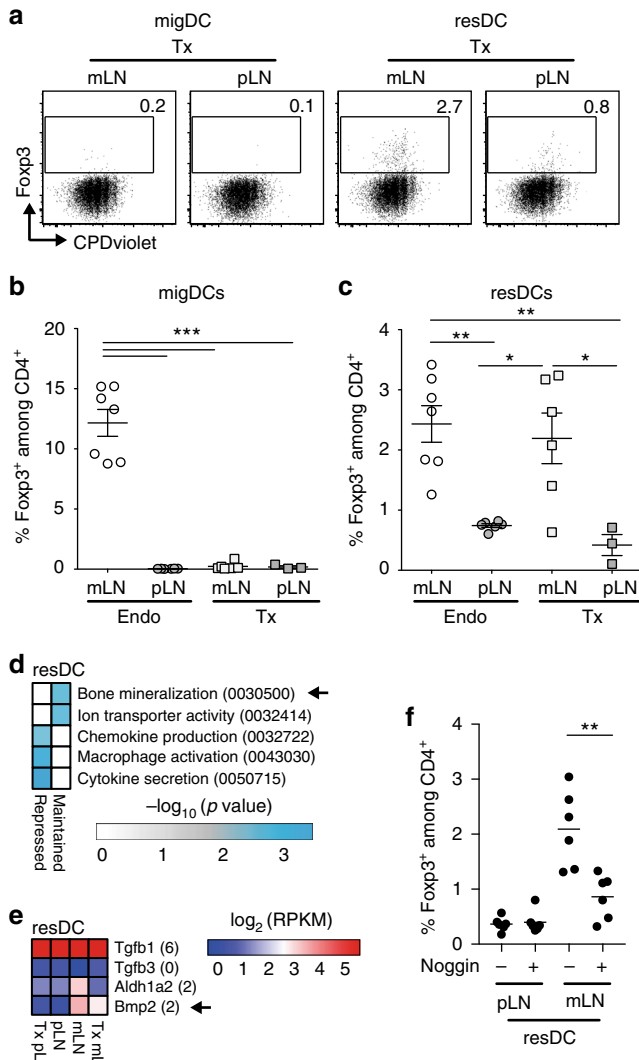

**Fig. 6** mLN SCs instruct resDCs with Treg-inducing properties. Indicated LNs were transplanted to the popliteal fossa of SPF-housed mice. Eight to sixteen weeks later, single cell suspensions were generated by enzymatic digestion, and respective DC populations isolated and co-cultured with naïve CPDviolet-labeled cells isolated from Foxp3[hCD2]xRag2[−/−]xDO11.10 mice and 160 ng/ml Ova$_{323-339}$ peptide for five days. **a** Representative dotplots of Foxp3 expression over CPDviolet dilution on gated CD4[+] T cells after co-culture with indicated DC subsets. Numbers indicate frequencies in gates. (**b–c**) Scatterplots show frequencies of Foxp3[+] Tregs among CD4[+] T cells after co-culture with migDCs (**b**) or resDCs (**c**) obtained from indicated locations. Data pooled from two to three independent experiments ($n = 3-7$). **d–e** resDCs were isolated from indicated LNs. RNA-seq[L] and subsequent analysis was performed. **d** GO analysis of biological processes of genes persistently up-regulated (Maintained) or down-regulated (Repressed) in resDCs isolated from transplanted mLN-SPF. **e** Heatmap of expression of indicated genes. Numbers in brackets indicate average $\log_2$(RPKM) expression of respective genes across all experimental groups. **f** Scatterplots show frequencies of Foxp3[+] Tregs among CD4[+] T cells after co-culture with resDCs obtained from indicated locations with or without supplementation of exogenous Noggin (100 ng/ml). Data pooled from two to three independent experiments ($n = 3-7$). migDC, migratory dendritic cell (Lin[−]CD11c[+]MHCII[high]); resDC resident dendritic cell (Lin[−]CD11c[high]MHCII[+]); Lin CD3[+]CD45R[+]Ly6G[+]F4/80[high], Endo endogenous, Tx transplanted

induced Foxp3$^+$ Tregs in the presence of mLN resDCs (Supplementary Fig. 7B). Together, these results suggest that mLN SCs can directly modulate resDCs and instruct them with Treg-inducing properties, thereby substantially contributing to the maintenance of intestinal tolerance.

## Discussion

Microbiota contribute to the acquisition of unique properties within the intestinal immune system, including the imprinting of a high Treg-inducing capacity in mLNs[11]. Here, we identify a critical time period for this imprinting process, delineate which non-endothelial SC subsets compile imprinted functional properties, and provide evidence that mLN SCs can shape functional properties of resDCs to attain a high Treg-inducing capacity.

Building on our previous observation that mLN SCs display stable tolerogenic properties[11], we here challenged mLN SCs concerning their tolerogenic function by exacting infectious and inflammatory perturbations. Infecting mice with *Y. pseudotuberculosis* resulted in the strong activation and proliferation of FSCs, in line with previous studies[21,24]. Remarkably, the tolerogenic properties of mLN SCs were not affected by this infectious perturbation, even though *Y. pseudotuberculosis* infection had been reported to propagate long-lasting, microbiota-dependent inflammation of mesenteric adipose tissue, finally resulting in an abrogation of mLN-dependent oral tolerance[44]. Moreover, chronic intestinal inflammation induced by repetitive DSS treatment failed to abrogate the high Treg-inducing capacity of mLNs, consistent with a recent report demonstrating a high topological robustness of the FSC network[45]. Together, these findings imply that the microbiota-dependent imprinting of tolerogenic properties within mLN SCs provides a very stable microenvironmental framework for efficient de novo Treg induction resistant to strong inflammatory perturbations.

Importantly, we identified the neonatal phase as the critical time period for the microbiota-dependent imprinting of the high Treg-inducing capacity within mLN SCs. During this phase, early after birth, the intestinal mucosa is rapidly colonized with microbiota, which has a profound impact on the development and maturation of a fully functional mucosal and systemic immune system[1–4]. Interference with the microbiota composition by antibiotic treatment revealed that a low-diversity microbiota was sufficient to imprint tolerogenic properties within mLN SCs. Remarkably, microbiota with an elevated proportion of *Lactobacillaceae*, prominent early colonizers of the intestine[46], fostered the imprinting of an even higher Treg-inducing capacity within mLN SCs, implicating evolutionarily conserved mechanisms to instill tolerance at the priming site of intestinal immune responses. It is important to note that the microbiota-dependent imprinting of tolerogenic properties within mLN SCs is not merely restricted to the neonatal phase as colonization of adult GF mice also resulted in stable Treg-inducing properties within mLNs, implying that mLN SCs retain functional flexibility until encountering microbiota-derived signals. These findings contrast with other microbiota-dependent tolerogenic mechanisms known to operate solely during the neonatal phase but not in adults[1,3,4].

Surprisingly, microbiota-derived signals scarcely impacted the transcriptomes of FSCs. In accordance with previous comparative microarray analyses[23,24], a large number of DEGs were identified between FSCs from pLNs and mLNs. Together, these data suggest that anatomic localization mainly determines the transcriptomes of FSCs rather than colonization status. Nonetheless, microbiota-derived signals were essentially required to stabilize the unique mLN-specific transcriptional signature within FSCs as evidenced by our finding that FSCs from transplanted mLN-GF were strongly influenced by the skin microenvironment and almost completely adapted to the transcriptional signature of FSCs from pLNs, while FSCs from transplanted mLN-SPF maintained a part of the mLN-specific transcriptional signature. The power of scRNA-seq[47] allowed attribution of these persistently up-regulated, as well as down-regulated genes to selected FSC subsets. Importantly, only a few of the fourteen transcriptional clusters identified here were known before[15], and a number of FSC subsets harboring unique immunomodulatory profiles are described here for the first time.

It is widely accepted that FSCs of secondary lymphoid organs regulate lymphocyte compartmentalization through chemokine secretion, form distinct niches for interaction with hematopoietic cells, and participate in the orchestration of appropriate cell-cell interactions required for adaptive immunity[15]. Splenic SCs can drive DC development from hematopoietic progenitors[48], maintain DC homeostasis[49], and support the differentiation of mature DCs to attain a tolerogenic phenotype[50]. Recently, FSCs from mLNs were shown to produce IL-15 for the maintenance of group 1 innate lymphoid cells[51], and that FSCs from LNs can function as niche cells, supporting the Notch-mediated differentiation of DCs, as well as follicular helper T cells[52]. In the present study we demonstrate that mLN SCs, once imprinted by microbiota, can directly modulate resDCs and instruct them with Treg-inducing properties. DCs are essential cellular players during the peripheral de novo generation of Foxp3$^+$ Tregs[11]. Previous studies have demonstrated that within the lamina propria migDCs can acquire the unique capacity to produce RA via direct crosstalk with lamina propria SCs in a RA- and GM-CSF-dependent manner[53], suggesting the functional education of migDCs within the intestinal tissue[54] which leads to the high Treg-inducing properties of CD103$^+$ migDCs within mLNs[9,10,55]. Interestingly, in the LN transplant setting we observed that skin-derived migDCs do not attain Treg-inducing properties when entering the transplanted mLN-SPF, suggesting that mLN SCs are incapable of reprogramming the functional properties of skin-derived migDCs. In contrast, resDCs could be efficiently modulated by SCs from transplanted mLN-SPF, as evidenced by subset composition, transcriptional signature and Treg-inducing properties. While the molecular mechanisms underlying this modulation still remain enigmatic, potential candidate genes could be deduced from the mLN-specific transcriptional signature stably maintained within FSCs from transplanted mLN-SPF. The RA-synthesizing enzymes *Aldh1a2* and *Aldh1a3*, showing a mutually exclusive, persistent expression within two distinct subsets of the adventitial/capsular CD34$^+$ SCs[28], are prime candidates as RA had been shown to modulate functional properties of DCs[54]. Although we were able to confirm that Aldh1a2/3 expression is higher in DCs from mLNs as compared to pLNs, defining the precise localization of the adventitial CD34$^+$ SCs expressing Aldh1a2/3 proved elusive, due to the unspecific binding of the applied detection antibody and the higher autofluorescence in the medullary areas of the LN. Other methods such as NICHE-seq or mRNA-based microscopy would thus be more appropriate[56,57].

When analyzing the transcriptional signature of resDCs that had been modulated by mLN SCs, several genes showing persistent up-regulation were identified. Among these Bmp2 was shown to act in concert with TGF-β in inducing Tregs[43]. As blocking of Bmp2-mediated signaling via Noggin abrogated mLN resDC-dependent Treg induction, mLN SC-dependent DC modulation could contribute to the maintenance of the tolerogenic properties of resDCs within mLNs, by elevating the Treg induction potential of resDC instilling higher Bmp2 expression.

In summary, we showed that microbiota stably and durably imprint tolerogenic properties within mLN SCs in the neonatal phase, rendering these resistant to inflammatory perturbations

later in life. Once stably imprinted, selected subsets of mLN SCs can mould the tolerogenic properties of resDCs, allowing these to efficiently induce Foxp3$^+$ Tregs de novo. Thus, crosstalk between mLN SC subsets and resDCs provides a robust feedback mechanism for the maintenance of intestinal tolerance.

## Methods

**Mouse strains**. Foxp3$^{hCD2}$xRag2$^{-/-}$xDO11.10 (BALB/c), Foxp3$^{hCD2}$x CD90.1 (BALB/c), CD45.1 (C57BL/6 and BALB/c), CD45.2 (BALB/c), CD90.1 (BALB/c), CD90.1 (C57BL/6), Actβ-eGFP (C57BL/6), Actβ-CFP (C57BL/6), and BALB/c mice from Janvier were bred and/or kept under SPF conditions in isolated ventilated cages at the Helmholtz Centre for Infection Research (Braunschweig, Germany). GF mice (BALB/c) were generated at Hannover Medical School (Hannover, Germany) by cesarean section and maintained either in plastic film isolators or in static micro-isolators (gnotocages) at Hannover Medical School or the Helmholtz Centre for Infection Research (Braunschweig, Germany). If not stated otherwise, water and Ova-free diet were supplied ad libitum. In all experiments, gender- and age-matched mice were used. All mice were housed and handled in accordance with good animal practice as defined by FELASA and the national animal welfare body GV-SOLAS. All animal experiments were approved by the Lower Saxony Committee on the Ethics of Animal Experiments, as well as the responsible state office (Lower Saxony State Office of Consumer Protection and Food Safety) under the permit number 33.9-42502-04-12/1012.

*Ccl19$^{Cre}$* mice[30] were purchased from EMMA (INFRAFRONTIER). *Rosa26$^{tdT}$*[31] and *Zbtb46$^{GFP}$* mice[32] were obtained from JAX. Mice were bred and maintained under specific-pathogen-free conditions at the Centre d'Immunologie de Marseille Luminy (Marseille, France). Experimental procedures were conducted in accordance with French and European guidelines for animal care under the permission numbers 5-01022012 following review and approval by the local animal ethics committee in Marseille (France).

**Antibodies**. The following fluorochrom-conjugated antibodies were used:

Anti-human CD2 (clone RPA-2.10; PE at 1:200 BioLegend Cat. 300208; FITC at 1:800 BioLegend Cat. 300214; PE-Cy7 at 1:400 BioLegend Cat. 309214; PerCP-Cy5.5 at 1:400 BioLegend Cat. 300216), anti-CD3 (clone 17A2; APC at 1:500 eBioscience Cat. 17-0032-82; PerCP-Cy5.5 at 1:400 BioLegend Cat. 100218), anti-CD4 (clone RM4-5; Alexa700 at 1:1000 eBioscience Cat. 56-0042-82; V500 at 1:1000 BD Cat. 560782), anti-CD8a (clone 53-7.3; APC/Fire750 at 1:800 BioLegend Cat. 100765; BUV395 1:100 BD Cat. 563786), anti-CD11b (clone M1/70; PerCP-Cy5.5 at 1:1000 eBioscience Cat. 12-0112-82; PacificBlue at 1:1000 BioLegend Cat. 101224; BV510 at 1:400 BioLegend Cat. 101245), anti-CD11c (clone N418; PE-Cy7 at 1:1000 BioLegend Cat. 117318; BV605 at 1:1000 BioLegend Cat. 117334; PacificBlue at 1:800 BioLegend Cat. 117322), anti-CD19 (clone 6D5; APC at 1:400 BioLegend Cat. 115512), anti-CD24 (clone M1/69; FITC at 1:1000 eBioscience Cat. 11-0242-85; APC at 1:400 BioLegend Cat. 101814), anti-CD31 (clone 390, PE-Cy7 at 1:1000 eBioscience Cat. 25-0311-82), anti-CD45 (clone 30-F11; FITC at 1:2000 BioLegend Cat. 103108; APC at 1:400 BioLegend Cat. 103112), anti-CD45.1 (clone A20; BUV395 at 1:200 BD Cat. 565212), anti-CD45.2 (clone 104; PE-Cy7 at 1:400 BioLegend Cat. 109830), anti-CD45R (clone RA3-6B2; APC at 1:800 BioLegend Cat. 103212; PercP-Cy5.5 at 1:800 BioLegend Cat. 103236), anti-CD49b (clone DX5; APC at 1:800 BioLegend Cat. 108910; PerCP-Cy5.5 at 1:100 BioLegend Cat. 108916), anti-CD62L (clone MEL14; PerCP-Cy5.5 at 1:2000 BioLegend Cat. 104432), anti-CD103 (clone 2E7; PE at 1:400 BD Cat. 58877; PE-Cy7 at 1:100 BioLegend Cat. 121426; APC at 1:200 BioLegend Cat. 121414), anti-F4/80 (clone BM8; BV605 at 1:100 BioLegend Cat. 123133; APC a 1:100 BioLegend Cat. 123110), anti-gp38 (clone 8.1.1; PE at 1:1000 BioLegend Cat. 127408), anti-Ly6C (clone HK1.4; Alex488 at 1:1000 BioLegend Cat. 128022; PerCP-Cy5.5 at 1:400 BioLegend Cat. 128013; APC at 1:2000 BioLegend Cat. 128016), anti-Ly6G (clone 1A8; APC at 1:400 BD Cat. 560599; PerCP-Cy5.5 at 1:1000 BioLegend Cat. 127616), anti-MHCII (clone M5/114.15.2; Alexa700 at 1:500 BioLegend Cat. 107622; PacificBlue at 1:2000 BioLegend Cat. 107620), anti-Ova-TCR (clone KJ1.26; FITC at 1:400 eBioscience Cat. 11-5808-82), anti-Ter119 (clone TER-119; APC at 1:500 BioLegend Cat. 118516). Fluorochrom-conjugated streptavidin and 7AAD were purchased from eBioscience, BD and BioLegend and used at a dilution of 1:1000. For immunostaining and confocal imaging, purified anti-ALDH1A1 (polyclonal rabbit IgG, cross-reactive to ALDH1A2) and its corresponding isotype control were obtained from Abcam (ab23375 and ab37415, respectively). Specific rat antibody recognizing mouse Foxp3 (FJK-16 S clone) was purchased from eBiosciences. Antibodies directed against rabbit and rat IgG were respectively purchased from Biotium (20373) and Jackson Immunoresearch (712-606-153).

**Cell isolation and flow cytometry**. Single cell suspensions from LNs were prepared by disintegrating organs through a 30 μm sieve (Sysmex Partec). Cells were resuspended in PBS containing 0.2% bovine serum albumin (BSA, Sigma-Aldrich) before staining. Cell suspensions were incubated with PBS containing 0.2% BSA and 10 μg/ml anti-mouse CD16/CD32 antibody (BioXcell) for 5 min on ice. Live/dead discrimination was carried out utilizing LIVE/DEAD Fixable Dead Cell Stain (Invitrogen) according to the manufacturer's recommendations. Surface staining

was performed for 15 min on ice in PBS containing 0.2% BSA. Cells were washed, resuspended in PBS containing 0.2% BSA and measured at LSRII or LSR Fortessa flow cytometers with Diva software (BD Biosciences). Data were analyzed using FlowJo software (Tree Star).

**LN transplantations**. For transplantations into the popliteal fossa, BALB/c-recipient or C57BL/6-recipient mice were anesthetized with ketamine (WDT) and xylazine (CP Pharma), the skin of the popliteal fossa of the right hind leg opened, and the endogenous pLNs and surrounding fat tissue removed. pLNs or mLNs dissected from donor mice were placed into the popliteal fossa, and the cut sewn with absorbable suture (Catgut). Before subjecting to further experimental procedure, recipients were housed for a minimum of eight weeks to ensure restoration of the lymphatic and blood vessel system. For all experiments, successful engraftment of transplanted LNs into the popliteal fossa was assessed by footpad injection of 20 μl Patent V (25 mg/ml, Sigma-Aldrich) into CO$_2$-euthanized mice.

**T-cell isolation and adoptive transfer**. For adoptive transfer of cells from Foxp3$^{hCD2}$xRag2$^{-/-}$xDO11.10 mice, single cell suspensions were generated from spleens and LNs and transferred without cell sorting. Before transfer, cells were labeled with cell proliferation dye CPDviolet (Invitrogen), and approximately 3–10 × 10$^6$ cells injected in 100 μl PBS (ThermoFisher Scientific) i.v. per recipient mouse.

**Immunization**. For de novo Foxp3$^+$ Treg induction, 20 μg Ova$_{323-339}$ peptide was injected i.v. on two consecutive days, starting one day after adoptive transfer of cells from Foxp3$^{hCD2}$xRag2$^{-/-}$xDO11.10 mice. At indicated time points after immunization, cells were isolated from transplanted and endogenous LNs.

**Gastrointestinal infection**. Prior to infection with *Yersinia pseudotuberculosis* strain YPIIIΔCNFy[58], food was removed for 12 h. Subsequently, BALB/c mice were infected intragastrically with 10$^9$ bacteria (in 200 μl PBS) that had been cultivated for 12 h at 25 °C in LB medium (BD). At indicated time points p.i., the bacterial load within mLNs was determined using single cell suspension aliquots homogenized at 30,000 rpm for 30 s using a Polytron PT 2100 homogenizer (Kinematica) and serial dilutions of the homogenates plated on LB plates. Colony forming units (CFU) were counted and are given as CFU/organ. For transplantation, separate cohorts of mice were utilized at the indicated time points p.i.

**Chronic DSS colitis**. To induce a chronic DSS colitis, mice were treated for four cycles with 5% DSS (36–50 kDa, MP Biomedicals) in drinking water ad libitum for four days followed by ten days of normal drinking water. Changes in body weight were monitored over time. At the end point of the experiment, colon length and spleen weight were determined.

**Antibiotic treatment**. Antibiotics were added to the drinking water seven days post conception with vancomycin hydrochloride at 200 mg/l, streptomycin sulfate at 200 mg/l or polymyxin B sulfate at 100 mg/l (all Sigma-Aldrich). Antibiotic solutions were exchanged twice per week. Offspring continuously received antibiotics treatment until donation of mLNs at age of four to five weeks.

**Bone marrow chimeras**. Mice were γ-irradiated (7 Gy) from an X-Ray source and reconstituted with at least 5 × 10$^7$ total bone marrow cells from donor animals. Chimera individuals were kept at least 40 days previous to resection of mLNs and pLNs.

**Immunostaining and microscopy**. LNs were fixed in Antigen Fix (Microm Microtech) for 2 h, washed in 0.1 M phosphate buffer, and dehydrated overnight in 30% sucrose at 0.1 M phosphate buffer. Tissues were snap frozen and 35 μm frozen sections stained with the indicated antibodies in 0.1 M UltraPure Tris Buffer (ThermoFischer) containing 0.5% w/v BSA and 1% Triton X-100. Immuno-fluorescence confocal microscopy was performed using a Zeiss LSM 880 confocal microscope. Separate images were collected for each fluorochrome and overlaid to obtain a multicolor image. Final image processing was performed with Imaris software (Bitplane).

**16 S rDNA sequencing**. DNA extraction, amplification of the V1-2 region of the 16 S rDNA gene and bioinformatics processing were performed as previously described[59] with minor modifications. The initial PCR was performed for 20 cycles with the following primers: 27 F: 5′-AGAGTTTGATCMTGGCTCAG-3′; 338 R: 5′-TGCTGCCTCCCGTAGGAGT-3′

Barcodes, indexes and Illumina sequence adaptors were appended via two additional PCR reactions. Firstly, a barcode was added in 15-cycle PCR reaction using the following primers: F2:
5′–ACACTCTTTCCCTACACGACGCTCTTCCGATCT*TTATGC*CAAGAGTT TGATCMTGGCTCAG-3′; R2:
5′–AATGATACGGCGACCACCGAGATCTACACTCTTTCCCTACACGACG CTCTTCCGATCT–3′ (Exemplary barcode indicated in italic). In a second step, an

index was added in a 10-cycle PCR reaction using the following primers: F3: 5′–AATGATACGGCGACCACCGAGATCTACACTCTTTCCCTACACGACG CTCTTCCGATCT–3′; R3: 5′–CAAGCAGAAGACGGCATACGAGATC*GT GAT*GTGACTGGAGTTC–3′ (Exemplary index indicated in italic). For the bioinformatics processing, paired-end raw sequences were assembled[60], subsequently aligned (gotoh algorithm with reference database SILVA) and pre-clustered (diff = 2) using MOTHUR. Obtained phylotypes were filtered for an average abundance of ≥ 0.001% of all samples. Annotation was performed as previously described[59]. Non-metric multidimensional scaling analysis (NMDS) is based on Bray-Curtis similarity on the phylotype level using PRIMER. Shannon index was calculated with R's package *phyloseq*[61].

**SC isolation**. For SC isolation, skin-draining pLNs (inguinal and axillary or popliteal) or mLNs (small intestinal and colon/caecum-draining) were resected and digested in RPMI 1640 medium (Gibco) containing 0.2 mg/ml collagenase P (Roche), 0.15 U/ml dispase (Roche) and 0.2 mg/ml DNase I (Roche) as described previously[23]. After digestion, cells were kept at 4 °C in PBS containing 0.2% BSA and 5 mM EDTA (Roth). CD45$^-$ cells were enriched by autoMACS separation after magnetic labeling of CD45$^+$ cells using anti-CD45-APC (30-F11, eBioscience) followed by anti-APC microbeads (Miltenyi Biotec) or anti-CD45 Nanobeads (Biolegend). Subsequently, the CD45$^-$ fraction was stained using fluorochrome-coupled antibodies and either analyzed directly by flow cytometry or used to sort CD45$^-$CD24$^-$CD31$^-$gp38$^+$ FSCs (Aria II, 100 μm nozzle) and bulk CD45$^-$CD24$^-$ non-hematopoietic cells (Aria III, 70 μm nozzle) for RNA-seq/RNA-seq$^L$ and scRNA-seq, respectively.

**DC isolation**. To isolate DC subsets, transplanted or endogenous pLNs and mLNs were excised and digested as described above. After digestion, cells were kept at 4 °C in PBS containing 0.2% BSA and 5 mM EDTA. For RNA-seq$^L$ analysis, the CD45$^+$ fraction of the above-mentioned autoMACS-separated cells were stained using fluorochrome-coupled antibodies and sorted for CD11c$^{high}$MHCII$^+$ resDCs and CD11c$^+$MHCII$^{high}$ migDCs[62,63] by FACS (Aria II, 100 μm nozzle). For the Treg-induction assay, lineage negative (CD3$^-$CD19$^-$CD49b$^-$) cells were enriched by autoMACS separation after magnetic labeling of lineage positive cells using FITC-conjugated anti-CD3, anti-CD19 and anti-CD49b antibodies followed by labeling with anti-FITC microbeads (Miltenyi Biotec). Subsequently, cells were labeled with fluorochrome-conjugated antibodies and sorted for resDCs and migDCs by FACS as described above.

**In vitro Treg-induction assay-DCs**. For the co-culture of DC subsets and naive CD4$^+$ T cells, single cell suspensions were generated from spleens and LNs of Foxp3$^{hCD2}$xRag2$^{-/-}$xDO11.10 mice, followed by enrichment of CD4$^+$ T cells using CD4 (L3T4) MicroBeads and autoMACS separation. CD4$^+$ T cells were then labeled with cell proliferation dye CPDviolet, and 2.5 × 10$^3$ cells plated into 96-well conical-bottom plates (Sarstedt) in RPMI 1640 medium supplemented with 10% FCS, 25,000 U penicillin, 25 mg streptomycin, 1 mM sodium pyruvate, 25 mM HEPES, 50 μM β-mercaptoethanol. All cultures were supplemented with 10 ng/ml IL-2 (R&D) and 160 ng/ml Ova$_{323-339}$ peptide. Additionally, 5 × 10$^2$ resDCs or migDCs were directly FACS-sorted to the wells as described above. Where indicated, co-cultures were supplemented with 250 ng/ml murine Bmp2 (R&D) or 100 ng/ml murine Noggin (BioLegend). At day 5, cells were resuspended, labeled with fluorochrome-conjugated antibodies, and the frequency of Foxp3$^{hCD2+}$ cells among CD4$^+$ T cells determined by flow cytometry.

**In vitro Treg-induction assay-SCs**. For the co-culture of FSCs from mLNs and pLNs with naive CD4$^+$ T cells, 96-well conical-bottom plates (Sarstedt) were coated with 0.25 mg/ml Fibronectin (Roche) at 37 °C for 1 h. 2.5 × 10$^3$ CD24$^-$CD45$^-$CD31$^-$gp38$^+$ FSCs were directly sorted into 150 μl of X-VIVO15 (Lonza) and cultured for one day. Subsequently, single cell suspensions were generated from spleens and LNs of Foxp3$^{hCD2}$xRag2$^{-/-}$xDO11.10 mice, followed by enrichment of CD4$^+$ T cells using CD4 (L3T4) MicroBeads and autoMACS separation. Next, CD4$^+$ T cells were labeled with cell proliferation dye CPDviolet. 2 × 10$^4$ CD4$^+$ T cells and 1 × 10$^4$ T-Activator CD3/CD28 Dynabeads were added in 50 μl of X-VIVO15. All cultures were supplemented with 10 ng/ml IL-2 (R&D) with or without 0.1 ng/ml TGFß1 and 2.5 μg/ml αTimd4 (RMT4-54). At day 4, cells were resuspended, labeled with fluorochrome-conjugated antibodies, and the frequency of Foxp3$^{hCD2+}$ cells among CD4$^+$ T cells was determined by flow cytometry.

**Library preparation RNA-seq**. Total RNA was extracted from FACS-sorted FSCs using the AllPrep DNA/RNA kit (Qiagen). Quality and integrity of total RNA were controlled on Agilent Technologies 2100 Bioanalyzer. Purification of poly A-containing mRNA was carried out via poly T oligo attached magnetic beads. Following purification, the mRNA was subjected to library preparation using Script Seq v2 Library preparation kit (Epicenter). Sequencing was carried out on Illumina HiSeq2500 using 50 bp single reads. Sequenced libraries were assessed for read quality using the FastQC tool and clipping performed using default settings of *fastq-mcf*. The Gene Expression Omnibus (GEO) accession number for the RNA-seq data reported in this paper is GSE116633.

**Library preparation low-input RNA-seq**. Total RNA was extracted from FACS-sorted FSCs and resDCs using the RNeasy Plus Micro Kit (Qiagen). cDNA was synthesized and amplified using template switching technology of the SMART-Seq v4 Ultra Low Input RNA Kit (Clontech Laboratories), followed by purification using the Agencourt AMPure XP Kit (Beckman Coulter). Library preparation was performed with Nextera XT DNA Library Prep Kit (Illumina). The Agilent Technologies 2100 Bioanalyzer was used to control quality and integrity of nucleic acids after each step. Deep sequencing was carried out on Illumina HiSeq2500 using 50 bp single reads. Sequenced libraries were assessed for read quality using the FastQC tool. The GEO accession number for the RNA-seq$^L$ data reported in this paper is GSE116633.

**Library preparation scRNA-seq**. Single cells were sorted by FACS ARIA III sorter (BD) and collected in PBS containing 0.04% w/v BSA at a densitiy of 400 cells/μl. Chromium$^{TM}$ Controller was used for partitioning single cells into nanoliter-scale Gel Bead-In-EMulsions (GEMs) and Single Cell 3′ reagent kit v2 for reverse transcription, cDNA amplification and library construction (10xGenomics). The detailed protocol was provided by 10xGenomics. SimpliAmp Thermal Cycler was used for amplification and incubation steps (Applied Biosystems). Libraries were quantified by Qubit$^{TM}$ 3.0 Fluometer (ThermoFisher) and quality checked using 2100 Bioanalyzer with High Sensitivity DNA kit (Agilent). Sequencing was performed in paired-end mode (2 × 75 cycles) using NextSeq 500 sequencer (Illumina) to attain approximately 80,000 reads per single cell. The GEO accession number for the scRNA-seq data reported in this paper is GSE116633.

**RNA-seq/RNA-seq$^L$ analysis**. Libraries were aligned versus the mouse reference genome assembly GRCm38 using the splice junction mapper Tophat2 v1.2.0 with default parameterization[64]. Reads aligned to annotated genes were quantified with the htseq-count program[65] and determined read counts served as input to DESeq2[66] for pairwise detection and quantification of differential gene expression. RPKM (reads per kilobase of exon length per million mapped reads) values were computed for each library from the raw read counts. For scatterplots and heatmaps only genes with an annotated official *Gene Symbol* were included. Gene ontology analyses were performed using the R package *TopGo*[67]. The R packages *pheatmap* and *ggplot2* were used to generate heatmaps or scatterplots, respectively.

**scRNA-seq analysis**. Data were demultiplexed using Cell Ranger software (version 2.0.0) based on 8 bp 10X sample indexes and paired-end FASTQ files generated. The cell barcodes and transcript UMIs were processed as previously described[68]. Subsequently, read 2, which contains the transcript insert sequence, was aligned to mouse UCSC mm10 reference genome using STAR aligner with default parameters[68]. The alignment results were used to quantify the expression level of mouse genes and generation of gene-barcode matrix, which were further processed with Seurat (version 2.3.1)[69]. The separately acquired mLN and pLN datasets were merged. All cells with less than 250 or more than 3,500 detected genes per cell were filtered out. Moreover, cells with more than 4.5% read mapping to mitochondrial genes were considered as dead cells and removed. The remaining cellular transcriptomes were normalized, log transformed and only cell clusters (non-endothelial SC) with expression for *Pecam* < 1 were used for the further analysis. Non-endothelial SC transcriptomes were regressed against the variables, number of UMIs and percent of mitochondrial reads. From the two independent experiments 2786 and 4483 non-endothelial SCs were obtained for mLNs and pLNs, respectively. Variable genes were identified with default settings. Based on the PCElbowPlot, 22 and 18 PCAs were used for the clustering of mLNs and pLNs, respectively. Clusters were defined at the resolution of 1.3 and 0.9 for mLNs and pLNs, respectively and DEGs calculated via FindAllMarkers (Wilcoxon rank sum test, min.pct = 0.25, thresh.use = 0.25, only.pos = TRUE). Projection of transcriptional signatures from mLN SC clusters to pLN SC clusters was carried out using *scmap*[29] with 1000 features and threshold set to 0.7.

To perform diagonal canonical correlation analysis (CCA)[69], mLNs and pLNs were additionally regressed against percent of ribosomal reads on a per cell basis, data sets were merged and the union of the variable genes was used. t-SNE dimensionality reduction was performed using the first 22 dimensions of the CCA and resolution set to 1.1. DEGs were identified as stated above.

Gene ontology (GO) analysis was performed for differentially upregulated genes per cluster using *TopGO*[67]. Cumulative Z-scores were calculated based on the scaled expression per gene per cluster across the defined gene signatures.

**Statistical analysis**. Group sizes were estimated according to a presumed standard deviation (SD) and an expected type I error of <0.05. The sample size was adjusted, if required, based on initial results. For all figures, each data point represents a single mouse if not stated otherwise. In Treg induction assays, samples were excluded from the statistical analysis if <500 adoptively transferred OvaTCR$^+$CD4$^+$ cells could be acquired. For comparison of unmatched groups, two-tailed Mann-Whitney was applied. If comparing more than three groups one-way ANOVA followed by Bonferroni's post-test was used. All data are presented as mean or mean ± SD, and *p*-values <0.05 deemed significant (*$p < 0.05$; **$p < 0.01$; ***$p < 0.001$; ****$p < 0.0001$; ns, not significant). Prism software (GraphPad) was utilized

for all flow cytometry based data. A Life Sciences Reporting Summary for this paper is available online.

## Data availability

The GEO accession number for all RNA-seq, RNA-seq$^L$ and scRNA-seq data reported in this paper is GSE116633.

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

## Acknowledgements

We thank Lothar Groebe for cell sorting, Robert Geffers for RNA-seq analysis, Daniel Hebenstreit for advice on RNA-seq data analysis, Fabio Pisano for support during infection experiments, Rainer Glauben for advice on DSS colitis experiments, Himpriya Chopra for technical fluorescence microscopy support, Vincent Gardeux for advice on scRNA-seq analysis, Jana Niemz for critical reading of the manuscript and Maria Ebel and Silke Kahl for technical assistance. This work was supported by the Hannover Biomedical Research School (HBRS), the Center for Infection Biology (ZIB) from Hannover Medical School, the German Research Foundation (SPP1656, Ho2236/9-2, PE 2840/1-1) and the European Union's Horizon 2020 research and innovation programme under the Marie Skłodowska-Curie grant agreement No 656319. The Helmholtz Institute for RNA-based Infection Research (HIRI) supported this work with a seed grant through funds from the Bavarian Ministry of Economic Affairs and Media, Energy and Technology (Grant allocation nos. 0703/68674/5/2017 and 0703/89374/3/2017).

## Author contributions

J.P., M.P., M.Z, C.W., M.B., G.-R.T., E.V., S.F., P.A., M.B., J.S., D.F., and M.V. performed experiments and interpreted data. D.-H.P., P.D., T.S., M.H., and U.B. provided expertize and feedback. M.B. and A.B. provided reagents. J.P., M.P., S.F., O.P., M.B., A.-E.S., and J. H. designed research, interpreted data, and wrote the manuscript.

## Additional information

**Competing interests:** The authors declare no competing interests.

