## [Peer Review File · Nature Communications]

Reviewers' comments:

Reviewer #1 (Remarks to the Author):

Previously, the Huehn group has shown that mesenteric lymph node (mLN) stromal cells have intrinsic properties favoring adaptive regulatory T-cell differentiation that are maintained upon transplantation in a non-mucosa draining anatomical location. Microbial colonization of the host was essential to maintain these transplantation resistant tolerogenic properties as transplanted mLN of germfree mice had no capacity to favor Treg induction.

In this manuscript Pezoldt et al. further dissect the role of microbial triggers in imprinting these tolerogenic stromal cell intrinsic properties. They show that mLN from recolonized germfree mice regain their tolerogenic properties, and that imprinting of the tolerogenic phenotype in SPF mice occurs between 0 and 10 days after birth. The transplantation resistant imprinting is long lasting as transplanted lymph nodes remain tolerogenic up to 50 weeks after transplantation. Moreover, the properties are resilient as antibiotic modulation of microbial composition does not impair imprinting and neither does gastrointestinal infection or chronic colitis. However, the mechanism responsible for the microbiota induced stromal imprinting remained elusive as transcriptomes from a subgroup of mLN stromal cells, the FRC, from SPF and germfree mice yielded negligible differences. Interestingly, the authors show that when the germfree mLN is in its original location it appears to compensate for the stromal defect and favors Treg differentiation to the same extent as an SPF mLN.

Altogether, this manuscript has a strong focus on the conditions that determine Foxp3 positive T-cell differentiation in a mucosal stromal niche. It emphasizes the extent to which stromal cells can modulate tolerogenic immune responses and sheds new light on microbial regulation of this process. Overall the conclusions are supported by the data, the data are well presented and the manuscript reads clearly. However, several findings raise questions.

Figure 1: Based on the previous data it was expected that the transcriptomes of FRC from germfree versus SPF mice would yield differentially expressed genes that relate to tolerogenic imprinting. The authors find 7 genes to be different. It is not clearly explained what these 7 genes are and why they were not followed up? Would the results have yielded more hits if the FRC had been sorted from the LN after injection of DO11.10 cells at the time of antigen presentation?

Figure 2: when the germfree mLN is in its original location Foxp3+ T-cell differentiation occurs to the same extent as an in an SPF mLN arguing that microbiota are dispensable for an efficient Treg induction in mLN. This appears counterintuitive but apparently other cellular processes in the mLN compensate for the lack of imprinting in the stroma. This lack of imprinting only becomes apparent when the lymph node is transplanted. Would this mean that the threshold for loss of oral tolerance is lower in a germfree mouse compared to SPF and could this be addressed in more detail? In other words: when would this compensation become insufficient and would the lack of stromal cell imprinting cause pathology?

Figure 3: a germfree control group would add to the strength of the figure. The range in Foxp3 frequency is quite variable. What explains this variation?

Figure 5: it is interesting that vancomycin treatment appears to boost Foxp3+ T cell differentiation. A significant change in microbiota composition amongst which Lactobacilli correlated with this. In view of the fact that Figure 1. yielded few candidate genes, would it be possible to assess whether lactobacilli imprint the transplantation resistant tolerogenic capacity of mLN stroma?

Reviewer #2 (Remarks to the Author):

In their manuscript Pezoldt et al. reports on the role of microbiota in imprinting stable tolerogenic properties of mLN stromal cells in neonates. These studies follow up on previous work in which the authors show that stromal cells are influenced by microbial signals within the mLN to promote regulatory T cell response (Cording et al. 2014). In this study, the authors use a similar approach but present data that to some degree contradict these previous finding that stromal cells within the mLN are highly influenced by the presence of the microbiota, but instead appear to be more influenced by anatomical location. As a result, the data presented are neutral or inconclusive and lack novelty.

Major concerns:

Figure 1: Localization, but not colonization status has strongest impact on transcriptome of FRC. How do the authors reconcile this result with their previous work? What are the 7 genes upregulated when comparing mLN SPF and GF?

Figure 2: Similarly, the authors show that microbiota is not necessary for an efficient Treg induction in the mLN and pLN by analyzing *foxp3*⁺ Treg frequency after OVA challenge in GF mice. This result seems to be contradictory with previous experiment showing that transplanted mLN taken from GF mice into popliteal fossa of SPF mice have significant reduced frequency of *foxp3*⁺ Tregs and from which the authors stated that microbiota is required for imprinting mLN stromal cells with tolerogenic properties. The authors argue that microbiota is not a prerequisite for an efficient Treg induction in mLN as long as the mLN are located in their original site. However, the authors didn't identify the mechanism that could explain such a discrepancy in their results. This is confusing why the authors didn't perform the analysis shown in Figure 1 with transplanted mLN from GF mice in popliteal fossa compared to SPF mice.

Figure 3: The authors show that mLN from conventionalized GF mice transplanted into popliteal fossa of SPF mice does not have reduced frequency in *foxp3*⁺ Tregs. They state from this negative result that microbiota is imprinting stromal cells with tolerogenic properties referring to previously published experiments. Although it is already published, the GF control is essential in this experiment and should be provided.

Figure 4: The authors show that transplanted neonatal mLN from SPF have a reduced Treg inducing capacity compared to transplanted mLN from older mice. They state from this experiment that tolerogenic properties of mLNs are acquired early during ontogeny in a microbiota dependent manner. However, this statement is purely correlative.

Figure 2,3,4: The absolute number of cells (T cells, Tregs) should be provided.

Reviewer #3 (Remarks to the Author):

This manuscript investigates the characteristics of mesenteric LN stromal cells with a focus on their capacity to induce regulatory T cells to foreign antigen. They convincingly show that there is no difference in the Treg-inducing function of mLN when they are compared between germfree and SPF mice, including few transcriptional changes in the LN fibroblasts. The authors then revert to differences previously reported in *Mucosal Immunology* 2014 (Cording et al.) where they had seen an impact by microbiota on mLN transplanted into the popliteal site, and then use this experimental

system to investigate the development of the Treg-inducing function of mLN stroma, as well as the stability of this process, including during acute and chronic infection.

While the experiments performed are of high quality, this reviewer is of the opinion that the experimental system used for figures 3-8 to study the impact of the microbiota, namely the transplantation of mLN into the popliteal site, is very unphysiological, especially as microbiota have no measurable effect on iTreg induction in non-transplanted mLN (Fig.2). So it boils down to pure academic questioning on the stability of the imprinting of stromal cells in the transplantation setting with no evidence for such a role in the mesenteries. In addition, it remains unclear what the relative roles are of microbiota versus food antigen in this process. As a consequence some conclusions in the abstract are not sufficiently supported by the experimental data. In my opinion, this work is therefore not suitable for this level of journal.

REDACTED

Reviewers' comments:

Reviewer #1 (Treg induction)(Remarks to the Author):

This study entitled "Neonatally imprinted mesenteric lymph node stromal cell subsets induce tolerogenic dendritic cells" by Pezoldt et al. is a new manuscript that has substantial overlap with a previously submitted manuscript entitled "Microbiota imprint stable, inflammation resistant tolerogenic properties of mesenteric lymph node stromal cells in neonatal phase" (NCOMMS-16-12371). Both manuscripts are based on previous work showing that mesenteric lymph node (mLN) stromal cells have intrinsic properties favoring adaptive regulatory T-cell differentiation that are maintained upon transplantation in a non-mucosa draining anatomical location. Microbial colonization of the host was essential to maintain these transplantation resistant tolerogenic properties as transplanted mLN of germfree mice had no capacity to favor Treg induction.

In the current manuscript the authors describe that tolerogenic properties of mLN stromal cells are stably retained and inflammation resistant; are rapidly acquired after birth and address the role of microbial triggers in imprinting these tolerogenic stromal cell intrinsic properties. The authors have added more in depth analyses of the signature of mLN from SPF colonized versus mLN germ-free stromal cells, performed a detailed study of stromal FSC subsets in mLN and skin-draining pLN and demonstrate differences in frequency and function of resident dendritic cell populations in transplanted mLN and pLN from adult SPF mice.

Similar to the previous manuscript this manuscript has a strong focus on the conditions that determine Foxp3 positive T-cell differentiation in a mucosal stromal niche. It emphasizes the extent to which stromal cells can modulate tolerogenic immune responses and sheds new light on microbial regulation of this process. Compared to the previous manuscript the current study has a stronger focus on the molecular signature of the FSC.

Overall the results are clear and the new data are interesting and add to the novelty of the manuscript. However, because of the large amount of data added there are some concerns regarding the presentation of the data and the clarity of the writing.

In the abstract the sentence "Utilizing LN transplantation, RNA-seq and single cell RNA-seq allowed identification of stably imprinted expression signatures in mLN fibroblastic stromal cells and dissected stromal cell subsets providing a niche for dendritic cell modulation." is too superficial and does not capture the data described in the manuscript.

---For example the paragraph in the results: "Remarkably, Aldh1a2, an enzyme responsible for RA synthesis, was persistently maintained only in FSCs from transplanted mLN-SPF but not in FSCs from transplanted mLN-GF, whereas the RA234 degrading enzyme Cyp26b1 remained repressed (Supplementary Fig. 3D).----- much more clearly describes the actual result. Could the authors better emphasize their results?"

The results section is sometimes difficult to follow as many of the data are placed in the supplementary figures.

For example summary of the most significant data in supplementary Figure 3 and supplementary Figure 5B should be included in Figures 3 and 5 of the main manuscript.

For figure 6 the overall rationale, experimental setup and results of figure 6 are clear. However, the interpretation of the effects seen in migratory DC warrant more caution. In this experiment LNs are transplanted to the popliteal fossa and resident versus migratory DC populations are isolated and their Treg inducing capacity is assessed in vitro. The authors observe that resident DC of transplanted mLN maintain their capacity to induce differentiation of Foxp3+ Treg. In the cocultures with purified migratory DC no preferential Foxp3+ Treg differentiation is observed. In the discussion the authors conclude: "Interestingly, in the LN transplant setting we observed that skin-derived migDCs do not

attain Treg-inducing properties when entering the transplanted mLN-SPF, suggesting that mLN stromal cells are incapable of 'reprogramming' the functional properties of skin-derived migDCs." It is questionable whether such a conclusion can be made on the basis of an in vitro experiment with DC's isolated from a "resting LN". Very few DC's constitutively migrate to a node in the popliteal fossa without antigenic stimulation of the skin and thigh muscle area. Migratory DC's obtained after subcutaneous injection as a trigger for migration may still yield different results.

Reviewer #3 (LN stroma)(Remarks to the Author):

Nature communications, Huehn et al. March 2018-03-12

In this manuscript, the authors continue their characterization of the tolerogenic function of fibroblastic stromal cells found within mesenteric but not peripheral lymph nodes, namely the induction of regulatory T cells by the stromal microenvironment (Cording et al., *Mucosal Immunology* 2014), by using LN transplantation between the mesenteric and the peripheral sites as main approach.

They convincingly demonstrate the importance of the neonatal time period for this stromal imprinting process on Treg, its stability over time and robustness towards inflammatory events, its dependence on microbiota but showing Ahr ligands and SCFA are insufficient. Then, the authors search for transcriptional differences in bulk fibroblastic stroma of mLN vs pLN by RNA seq., indicating anatomical differences which were maintained after transplantation but surprisingly were hardly impacted anymore by the microbiome. Interestingly, mLN from germfree mice were reprogrammable when transplanted in a skin-draining microenvironment pointing to a role for the microbiome in stabilizing the mLN transcriptional signature. To identify the stroma subset responsible for this Treg imprinting process, they did single cell RNAseq identifying a plethora of different cell subsets, many of which seem to be fibroblastic. Some of the key genes (*Aldh1a2/3*) responsible for Treg induction were found to be enriched in two smaller fibroblast subsets.

At this point, the manuscript switches and continues with the description of the impact of the stromal cell microenvironment on DCs that also have a critical role in Treg induction within mLN, including a demonstration that the transcriptome of resident DC is only partially maintained upon transplantation. Nevertheless, stroma of transplanted mLN had a higher propensity to induce Treg than those from pLN.

The experiments seem to be well done, the results novel and carefully analysed with sound statistical analysis and the conclusions well supported by the data. These results should be of interest to a wider immunology audience.

Half of this manuscript resembles a resource article (rich in interesting transcriptional data) but where the molecular pathway responsible for Treg induction has not yet been sorted out (both for stromal fibroblasts and for resident DCs tx data are provided), as elaborated below.

Major points:

- 1) Despite the wealth of molecular information contained within the sc RNAseq analysis of figure 4, the reader is left with a blurry definition of two small fibroblast subsets expressing *Aldh1* for which we do not know their anatomical localization, capacity to metabolize vitamin A/retinoic acid or propensity to induce Treg. While functional assays may be difficult to perform given the low cell number and difficulty to identify surface markers for cell sorting/enrichment, some further histological analysis would be informative on their anatomical localization. In situ hybridization (RNAscope) using two probes (*Aldh1* and *gp38* or *CD11c*) or histology using an RA reporter system (aldefluor) would allow to identify the microenvironment where these fibroblasts and DCs are found, and to see whether these

are the same microenvironments and whether Treg localize there.

2) Similarly, in figure 5E, S5 and 6 the reader wonders about a molecular pathway which could be responsible for the Treg inducing capacity of the resident DCs (we are left with another gene list...). Is it still Aldh1 in this context? Do inhibitors of the RA pathway block Treg induction in this vitro assay? Do fibroblasts (or gross subsets) of these LNs enhance the Treg induction in presence of resDC or migDC? Is GMCSF also produced by RA+ stromal cells given that this signal enhances the generation of RA+ CD103+ DCs (Vicente et al. Mucosal Immunology 2014)?

3) The single cell RNA seq experiment shown in figures 4 and S4 suggest 13 different fibroblast subsets and is a nice resource for future exploration. However, this reviewer is rather confused by the way subsets were attributed based it seems on a single positive marker: pericytes due to the acta2, MRC due to Madcam (CCL19 is not selective for MRC). Do other markers confirm this classification? MRC are expected to be also rankl+, CXCL13+; pericytes also NG2+. Several subsets could not be clearly attributed to a known cell type, microenvironment or a given function. Were FDCs not identified (which should be mfg8+CD35+ etc)? The scientific community would also appreciate an analysis where mLN and pLN cells are separated.

Minor points:

- 1) Line 128 may need rephrasing
- 2) ACKR4 antibody source is not listed

We highly appreciate the interest expressed by you and would like to thank you for your constructive critique and useful comments, which has helped us to improve the overall quality of our manuscript. Based on your valuable suggestions we have now incorporated several new experimental data and provided all relevant information in the revised version of the manuscript (all changes are highlighted in red).

In particular, we have 1) elucidated the molecular mechanism underlying the differential Treg induction capacity between resident DCs and migratory DCs; 2) performed additional scRNA-seq experiments; 3) extensively reanalyzed the data to better associate the proposed cell subsets with known cell types; and 3) performed immunostainings together with confocal imaging of tissue-sections prepared from lymph nodes to identify the localization of *Aldh1a2*-expressing cells.

Please find below our point-by-point response to your comments. We believe that we have addressed all concerns expressed by you and trust that our manuscript is now worthy of publication in Nature Communications.

Reviewer # 1 comments	Response by the authors
General comment #1: Overall the results are clear and the new data are interesting and add to the novelty of the manuscript. However, because of the large amount of data added there are some concerns regarding the presentation of the data and the clarity of the writing.	We highly appreciate the general statement of this reviewer regarding the clarity and novelty of our results. As outlined in detail below, we have made a number of attempts to optimize the presentation of the data and clarity of the writing.
Specific comment #1: In the abstract the sentence “Utilizing LN transplantation, RNA-seq and single cell RNA-seq allowed identification of stably imprinted expression signatures in mLN fibroblastic stromal cells and dissected stromal cell subsets providing a niche for dendritic cell modulation.” is too superficial and does not capture the data described in the manuscript. For example the paragraph in the results: “Remarkably, Aldh1a2, an enzyme responsible for RA synthesis, was persistently maintained only in FSCs from transplanted mLN-SPF but not in FSCs from transplanted mLN-GF, whereas the RA234 degrading enzyme Cyp26b1 remained repressed (Supplementary Fig. 3D). ... much more clearly describes the actual result. Could the authors better emphasize their results?”	We thank the reviewer for raising this point. We have emended the abstract to reduce its superficiality and to better emphasize the exciting results (see page 3).
Specific comment #2: The results section is sometimes difficult to follow as many of the data are placed in the supplementary figures. For example summary of the most significant data in supplementary Figure 3 and supplementary Figure 5B should be included in Figures 3 and 5 of the main manuscript.	We agree with this reviewer’s comments and have followed the recommendation to integrate datasets from the Supplementary Figures into the main manuscript to improve its readability and quality. The following integrations have been carried out:  • Supplementary Figure 1E-G to Figure 1E & F (see also page 7) • Supplementary Figure 3D & E to Figure 3D (see also page 11) • Previous Supplementary Figure 5B to Figure 5F (see also page 14&15)
Specific comment #3: For figure 6 the overall rationale, experimental setup and results of figure 6 are clear. However, the interpretation of the effects seen in migratory DC warrant more caution. In this experiment LNs are transplanted to the popliteal fossa and resident versus migratory DC populations are isolated and their Treg inducing	We fully agree with this reviewer that it is difficult to draw conclusions on the basis of in vitro experiments with DCs isolated from transplanted LNs. Accordingly, we performed a number of experiments to better characterize the migDCs in the transplanted LNs.

capacity is assessed *in vitro*. The authors observe that resident DC of transplanted mLN maintain their capacity to induce differentiation of Foxp3+ Treg. In the cocultures with purified migratory DC no preferential Foxp3+ Treg differentiation is observed. In the discussion the authors conclude: “Interestingly, in the LN transplant setting we observed that skin-derived migDCs do not attain Treg-inducing properties when entering the transplanted mLN-SPF, suggesting that mLN stromal cells are incapable of ‘reprogramming’ the functional properties of skin-derived migDCs.”

It is questionable whether such a conclusion can be made on the basis of an *in vitro* experiment with DC’s isolated from a “resting LN”. Very few DC’s constitutively migrate to a node in the popliteal fossa without antigenic stimulation of the skin and thigh muscle area. Migratory DC’s obtained after subcutaneous injection as a trigger for migration may still yield different results.

First, we transplanted LNs from CD45.1 congenic mice into the popliteal fossa of CD45.2 recipient mice to unravel the ‘chimerism’ of donor- and recipient-derived cells within the DC subsets of interest. We could demonstrate that at the time of analysis (eight to sixteen weeks after transplantation) both resDC and migDC subsets within the transplanted LNs were fully replaced by recipient-derived cells. These data, which can be found in the **novel Supplementary Fig. 6A** and the corresponding paragraph of the results section on **page 14**, demonstrate that migDCs, most likely originating from the skin, have migrated to the transplanted LN even under ‘resting’ conditions. Thus, we refrained from triggering DC migration by subcutaneous injections as suggested by this reviewer since we were afraid that the injection might result in an unwanted inflammatory perturbation abrogating the required steady-state conditions and also that we would not be able to re-isolate sufficient numbers of subcutaneously injected DCs from the transplanted LNs to perform subsequent *in vitro* experiments.

Second, we performed low-input RNA-seq analyses of migDCs isolated from endogenous and transplanted pLNs and mLNs to investigate to which degree migDCs can be modulated by pLN and mLN stromal cells. By comparing their transcriptional profiles, we conclude that:

- migDCs from endogenous mLNs and pLNs are transcriptionally more distinct than resDCs (**see Figure 5E and novel Supplementary Figure 6C**)
- a substantial fraction of these location-dependent DEGs (159 out of 1666) was stably maintained in migDCs isolated from transplanted mLNs (**see novel Supplementary Figure 6C**)
- migDCs isolated from transplanted LNs retained the expression of characteristic marker genes, such as *Ccr7*, *Arc* and *Irf4* (**see novel Supplementary Figure 6D**), providing further evidence that migDCs have migrated to the transplanted LN even under ‘resting’ conditions.

These novel findings, to which we refer in the results section on **pages 14&15**, demonstrate that mLN SCs can also modulate migDCs at the molecular level,

	albeit lacking any effect on their Treg-inducing properties.
--	--

Reviewer # 2 comments	Response by the authors
General comment #1: The experiments seem to be well done, the results novel and carefully analysed with sound statistical analysis and the conclusions well supported by the data. These results should be of interest to a wider immunology audience. Half of this manuscript resembles a resource article (rich in interesting transcriptional data) but where the molecular pathway responsible for Treg induction has not yet been sorted out (both for stromal fibroblasts and for resident DCs tx data are provided), as elaborated below.	We greatly appreciate the very positive statement of reviewer #1 regarding the novelty and quality of our results and their interest to a wider immunological audience. As outlined in detail below, we have performed a number of experiments to better characterize the molecular pathways underlining the differential Treg induction in mLNs and pLNs.
Major comment #1: Despite the wealth of molecular information contained within the sc RNAseq analysis of figure 4, the reader is left with a blurry definition of two small fibroblast subsets expressing Aldh1 for which we do not know their anatomical localization, capacity to metabolize vitamin A/retinoic acid or propensity to induce Treg. While functional assays may be difficult to perform given the low cell number and difficulty to identify surface markers for cell sorting/enrichment, some further histological analysis would be informative on their anatomical localization. In situ hybridization (RNAscope) using two probes (Aldh1 and gp38 or CD11c) or histology using an RA reporter system (aldefluor) would allow to identify the microenvironment where these fibroblasts and DCs are found, and to see whether these are the same microenvironments and whether Treg localize there.	We are grateful for the reviewer's appreciation regarding the extent of molecular information we provide. We have adapted the nomenclature for the newly identified stromal cell subsets also taking into consideration a very recently published study (Rodda et al. Single-Cell RNA Sequencing of Lymph Node Stromal Cells Reveals Niche-Associated Heterogeneity. Immunity 48, 1014-1028 e1016 (2018)). We agree with this reviewer that it is extremely challenging to perform functional assays with LN stromal cell subsets due to their paucity and the lack of (surface) markers enabling their isolation. Thus, we followed the reviewers' recommendation and tried to specify the anatomical localization of the Aldh1a2-expressing stromal cells. With the help of Marc Bajenoff, a renowned expert on stromal cell immunobiology, and his team we were able to confirm that LN stromal cells of the T cell zone do not express Aldh1a2, but that DCs are the major Aldh1a2-expressing cell subset within mLNs. Unfortunately, we were not able to reliably identify Aldh1a2-expressing stromal cells within the capsular region and in the vicinity of the medullary cords due to unspecific background signals of the utilized antibody and an increased autofluorescence. Importantly, Foxp3⁺ were equally dispersed in mLNs and pLNs within their T cell zones and did not show any obvious preferential interaction with Aldh1a2-expressing DCs. These data were integrated into the novel Supplementary Figure. 5 and we refer to the novel findings

	in the results section (page 13), in the discussion section (page 19&20), and in the methods section (page 21, 22, 24 & 25).
Major comment #2: Similarly, in figure 5E, S5 and 6 the reader wonders about a molecular pathway which could be responsible for the Treg inducing capacity of the resident DCs (we are left with another gene list...). Is it still Aldh1 in this context? Do inhibitors of the RA pathway block Treg induction in this vitro assay? Do fibroblasts (or gross subsets) of these LNs enhance the Treg induction in presence of resDC or migDC? Is GMCSF also produced by RA+ stromal cells given that this signal enhances the generation of RA+ CD103+ DCs (Vicente et al. Mucosal Immunology 2014)?	We thank the reviewer for emphasizing the requirement to better dissect the molecular mechanism underlying the enhanced Treg-inducing capacity of resDCs from mLNs. We performed a number of in vitro Treg induction assays and did not observe any direct impact of mLN stromal cells on naïve T cells during de novo Treg induction (see novel Supplementary Figure 7A and corresponding paragraph of the results section on page 15). Furthermore, we assessed if blocking of Timd4, highly expressed by a fraction of pLN stromal cells (data not shown), could abrogate Treg induction under suboptimal conditions (low TGFβ1 concentration). Yet, this was not the case (see novel Supplementary Figure 7A), suggesting that indeed DCs are required to confer the immune modulatory functions of mLN stromal cells. Closer examination of the RNA-seq data did not reveal any increased Aldh1a2 expression in resDCs isolated from transplanted mLNs as compared to resDCs isolated from transplanted pLNs (see novel Figure 6E). This led us to conclude that resDCs do not require Aldh1a2 expression to promote Treg induction. Therefore, we proceeded to assess whether Bmp2, a soluble protein described to synergize with TGFβ1 to promote Treg induction (Lu et al. Synergistic effect of TGF-beta superfamily members on the induction of Foxp3⁺ Treg. Eur J Immunol 40, 142-152 (2010)) might be involved in the increased Treg induction mediated by resDCs isolated from mLNs. First, resDCs isolated from transplanted mLNs showed an increased Bmp2 expression when compared to resDCs isolated from transplanted pLNs (see novel Figure 6D&E). Secondly, blocking of Bmp2 signaling via the antagonist Noggin ameliorated the increased Tregs induction mediated by resDCs isolated from mLNs (see novel Figure 6F). Finally, the increased Treg induction mediated by resDCs isolated from mLNs could be even slightly enhanced by addition of recombinant Bmp2, while Bmp2 did not show this synergistic effect on resDCs

	isolated from pLNs (see novel Supplementary Figure 7B). From these novel findings, to which we refer in the abstract (page 3) results section (page 15&16), in the discussion section (page 20), and in the methods section (page 26&27), we concluded that mLN stromal cells enable resDCs to express Bmp2, thereby elevating their Treg-inducing capacity. We thank the reviewer for pointing out the functional relevance of GM-CSF (Csf2) by stromal cells. Yet, within our scRNA-seq data we were not able to detect Csf2 expression, in line with the recently published study by Rodda et al. also using the 10X Genomics platform (Single-Cell RNA Sequencing of Lymph Node Stromal Cells Reveals Niche-Associated Heterogeneity. Immunity 48, 1014-1028 e1016 (2018); Link to gene expression platform: http://scorpio.ucsf.edu/shiny/LNSC/).
Major comment #3: The single cell RNA seq experiment shown in figures 4 and S4 suggest 13 different fibroblast subsets and is a nice resource for future exploration. However, this reviewer is rather confused by the way subsets were attributed based it seems on a single positive marker: pericytes due to the acta2, MRC due to Madcam (CCL19 is not selective for MRC). Do other markers confirm this classification? MRC are expected to be also rankl+, CXCL13+; pericytes also NG2+. Several subsets could not be clearly attributed to a known cell type, microenvironment or a given function. Were FDCs not identified (which should be mfge8+CD35+ etc)? The scientific community would also appreciate an analysis where mLN and pLN cells are separated.	We thank this reviewer for the critical comment on the way subsets were defined and on the recommendation to include additional markers in the scRNA-seq analysis. Indeed, we used a multitude of markers to identify known subsets like pericytes, but agree that these should be better highlighted in the manuscript to increase clarity. Therefore, we have included additional markers in the respective figures and the corresponding parts of the results section (see Figure 4 and pages 11-13). Furthermore, we provide extensive Supplementary Tables for all DEGs per subset, which we hope will benefit the scientific community to better dissect known and novel LN stromal cell subsets. Furthermore, we have included separate in-depth analysis of mLNs and pLNs and used scmap to project transcriptional signatures per subset from mLN stromal cells to pLN stromal cells in order to identify common subsets across different LNs (also see pages 29&30 in the methods section). We were also surprised to find any FDCs within our scRNA-seq data as no cell expressing Cr1 (CD35) nor Cr2 could be detected. Additionally, Ng2 could also not

	be detected. Mfg8 expression was found to be ubiquitously expressed by many stromal cell subsets (data not shown). We thus concluded that with the isolation protocol used for the present study we were not able to isolate FDCs from LNs under steady-state conditions. Regarding the definition of MRCs, we very much appreciate the feedback of the reviewer. Upon closer inspection, the MRC subset pronounced in the initial submission of the manuscript closely resembles the Madcam⁺Ccl19^{high} cells described in the study by Rodda et al. (Single-Cell RNA Sequencing of Lymph Node Stromal Cells Reveals Niche-Associated Heterogeneity. Immunity 48, 1014-1028 e1016 (2018)). In general, MRCs are thought to express Pdpr, Tnfrsf11, Vcam1, Icam1, Bst1, Relb, Madcam1 and Cxcl13. As can be seen from the Figure “Expression of putative key MRC genes” (see below) provided for review only, Tnfrsf11 is expressed significantly higher in Inmt⁺ stromal cells (see Supplementary Table 3), yet these cells lack pronounced Madcam1 and Cxcl13 expression. The propensity of Tnfrsf11⁺Cxcl13⁺ stromal cells to express Madcam1 sparingly was also observed by Rodda et al. (Single-Cell RNA Sequencing of Lymph Node Stromal Cells Reveals Niche-Associated Heterogeneity. Immunity 48, 1014-1028 e1016 (2018)). We thus conclude that to precisely define MRCs additional methods such as NICHE-seq (Medaglia et al. Spatial reconstruction of immune niches by combining photoactivatable reporters and scRNA-seq. Science 358, 1622-1626 (2017)) need to be employed, which we believe is outwith the scope of the present study.
Minor comment #1: Line 128 may need rephrasing.	We thank the reviewer for the feedback and have rephrased the sentence in question for clarity (see page 7).
Minor comment #2: ACKR4 antibody source is not listed.	We apologize for the misunderstanding, but Ackr4 expression was not assessed on the protein level. We have now emphasized in the manuscript that Ackr4 was detected at the level of mRNA.

Figure for review only “Expression of putative key MRC genes”: Scaled expression of markers associated with MRCs onto the t-SNE-map (for cluster nomenclature see Figure 4A).

REVIEWERS' COMMENTS:

Reviewer #1 (Remarks to the Author):

The authors have substantially revised the manuscript in response to the reviewers suggestions. The amendments have significantly improved the quality of the data and clarified the message of the manuscript.

Reviewer #3 (Remarks to the Author):

The authors have made the necessary efforts to resolve the outstanding points to the satisfaction of this reviewer. The story line of this manuscript is now much improved and the conclusions well supported by the data, and with greater depth. I congratulate the authors to this well performed and data-rich study. There are only few minor points that I ask the authors to address.

- Fig. 4E lacks the scale for the expression level which was shown in a previous version of the manuscript
- text line 130: This analysis indicated
- lines 259-261: the coverage level of transcripts with traditional sc-RNA analysis is rather low (typically around 10%); possibly the reader should be informed about it (in the methods) so that the data from low transcript level genes is interpreted with the necessary care. This may not apply to pdpn and pecam that are presumably expressed at high levels.
- line 303: according to the methods section this antibody recognizes both Aldh1a1 and Aldh1a2 and the text here should reflect this
- line 304: whereas no Aldh1a1/2-expressing SC
- line 369: please tone down this sentence as it seems overinterpreted (the in vitro data are not sufficient to firmly conclude how things work in vivo)
- discussion: assumes that all differences seen between SC or DC should be visible at the level of transcription; a small statement about other levels of regulation invisible by tx analysis should be added.
- discussion (line 426): here the discussion switches from FSC to SC (the latter includes also EC), and the reader could be made conscious of the fact that transplanted EC could also have a role in the described process in vivo (eg. LEC could also be imprinted and mediate DC entry in a differential way).
- line 559: were really 50mio cells transferred and not 10x less?
- the GEO-deposited data should include all data from the FSC and DC analysis so that the community does not need to ask the authors for those data

Reviewer # 3 comments	Response by the authors
General comment #1: The authors have made the necessary efforts to resolve the outstanding points to the satisfaction of this reviewer. The story line of this manuscript is now much improved and the conclusions well supported by the data, and with greater depth. I congratulate the authors to this well performed and data-rich study. There are only few minor points that I ask the authors to address.	We thank this reviewer for this very positive comment and also for the careful reading of our manuscript. A point-by-point response to the few minor points can be found below.
Minor comment #1: - Fig. 4E lacks the scale for the expression level which was shown in a previous version of the manuscript	We thank the reviewer for raising this point and apologize for this mistake. The corresponding scale has been added to Figure 4E.
Minor comment #2: - text line 130: This analysis indicated	We changed the text as suggested.
Minor comment #3: - lines 259-261: the coverage level of transcripts with traditional sc-RNA analysis is rather low (typically around 10%); possibly the reader should be informed about it (in the methods) so that the data from low transcript level genes is interpreted with the necessary care. This may not apply to pdpn and pecam that are presumably expressed at high levels.	We agree with the reviewers concern that currently available single-cell sequencing methods can only recover a fraction of the transcripts present within each cell. Therefore, we have applied stringent thresholds (min.pct = 0.25 instead of default min.pct = 0.1) whilst using FindAllMarkers function. Thus, we only report DEGs that could be identified in at least 25 % of the cells in one of the respectively compared cellular subsets. We already had indicated this procedure in the methods section.

Minor comment #4: - line 303: according to the methods section this antibody recognizes both Aldh1a1 and Aldh1a2 and the text here should reflect this	We thank the reviewer for raising this point and apologize for this mistake. The text was modified as suggested.
Minor comment #5: - line 304: whereas no Aldh1a1/2-expressing SC	We modified the text as suggested.
Minor comment #6: - line 369: please tone down this sentence as it seems overinterpreted (the in vitro data are not sufficient to firmly conclude how things work in vivo)	We rephrased the corresponding sentence as suggested by this reviewer.
Minor comment #7: - discussion: assumes that all differences seen between SC or DC should be visible at the level of transcription; a small statement about other levels of regulation invisible by tx analysis should be added.	We fully agree with this reviewer that transcriptomic analyses do not allow identifying all differences between the cell types analyzed in this study. Yet, in our opinion it is clear from the current manuscript that we have focused our study on the identification of differences at the transcriptional level. Thus, we have decided not to comment explicitly on this point in the discussion to avoid any confusion.
Minor comment #8: - discussion (line 426): here the discussion switches from FSC to SC (the latter includes also EC), and the reader could be made conscious of the fact that transplanted EC could also have a role in the described process in vivo (eg. LEC could also be imprinted and mediate DC entry in a differential way).	The abbreviations 'SC' and 'FSC' have been introduced already in the introduction, and used carefully throughout the whole manuscript.
Minor comment #9: - line 559: were really 50mio cells transferred	Indeed the number of adoptively transferred cells is higher than for most published BM chimera models. To ensure reconstitution

and not 10x less?	within six weeks, 5*10exp7 cells were used.
Minor comment #10: - the GEO-deposited data should include all data from the FSC and DC analysis so that the community does not need to ask the authors for those data	The GEO accession number for all RNA-seq, RNA-seq ^L and scRNA-seq data reported in this paper is GSE116633. https://www.ncbi.nlm.nih.gov/geo/query/acc.cgi?acc=GSE116633 We slightly modified the 'Data Availability' section to clarify that all transcriptomic data reported in this study are available for the community through the abovementioned link.